# PRPF8-mediated dysregulation of hBrr2 helicase disrupts human spliceosome kinetics and 5´-splice-site selection causing tissue-specific defects

Robert Atkinson[1,13], Maria Georgiou[1,13], Chunbo Yang[1,13], Katarzyna Szymanska[2,13], Albert Lahat[3,13], Elton J. R. Vasconcelos[4,13], Yanlong Ji[5,6], Marina Moya Molina[1,7], Joseph Collin[1], Rachel Queen[1], Birthe Dorgau[1], Avril Watson[1,7], Marzena Kurzawa-Akanbi[1], Ross Laws[8], Abhijit Saxena[1], Chia Shyan Beh[1], Chileleko Siachisumo[1], Franziska Goertler[9], Magdalena Karwatka[2], Tracey Davey[8], Chris F. Inglehearn[2], Martin McKibbin[2], Reinhard Lührmann[5], David H. Steel[1], David J. Elliott[1], Lyle Armstrong[1], Henning Urlaub[5,6,10], Robin R. Ali[11,14], Sushma-Nagaraja Grellscheid[3,9,14], Colin A. Johnson[2,14] ✉, Sina Mozaffari-Jovin[5,12,14] ✉ & Majlinda Lako[1,14] ✉

The carboxy-terminus of the spliceosomal protein *PRPF8*, which regulates the RNA helicase Brr2, is a hotspot for mutations causing retinitis pigmentosa-type 13, with unclear role in human splicing and tissue-specificity mechanism. We used patient induced pluripotent stem cells-derived cells, carrying the heterozygous *PRPF8* c.6926 A > C (p.H2309P) mutation to demonstrate retinal-specific endophenotypes comprising photoreceptor loss, apical-basal polarity and ciliary defects. Comprehensive molecular, transcriptomic, and proteomic analyses revealed a role of the PRPF8/Brr2 regulation in 5'-splice site (5'SS) selection by spliceosomes, for which disruption impaired alternative splicing and weak/suboptimal 5'SS selection, and enhanced cryptic splicing, predominantly in ciliary and retinal-specific transcripts. Altered splicing efficiency, nuclear speckles organisation, and PRPF8 interaction with U6 snRNA, caused accumulation of active spliceosomes and poly(A)+ mRNAs in unique splicing clusters located at the nuclear periphery of photoreceptors. Collectively these elucidate the role of PRPF8/Brr2 regulatory mechanisms in splicing and the molecular basis of retinal disease, informing therapeutic approaches.

The spliceosome is a multi-megadalton ribonucleoprotein complex that is composed of five small nuclear ribonucleoproteins (snRNPs) and >200 polypeptides that assemble on pre-mRNAs in a stepwise manner to catalyse intron excision[1]. The selection of specific exons/introns produce different transcripts from one genomic locus, leading to alternative splicing that is often regulated in a tissue-specific manner.

Splicing is initiated by interaction of the U1 and U2 snRNPs with the pre-mRNA 5' splice site (5'SS) and branch site, respectively, leading to the generation of pre-spliceosomes (complex A). The U4/U6.U5 tri-

snRNP then joins, leading to the formation of the pre-activated spliceosome complex B, where U1 has been released (by the action of PRPF28) and 5′SS transferred to the conserved ACAGA-box of the U6 snRNA. The B-complex undergoes profound compositional and conformational rearrangements, including the dissociation of U4/U6 di-snRNP. The release of U4, by the action of Brr2 (SNRNP200), a structurally exceptional spliceosomal RNA helicase, allows free U6 to

form the spliceosome catalytic centre, which yields the activated spliceosome (B[act])[2].

Pre-mRNA processing factors (PRPFs), including the large highly conserved scaffolding protein PRPF8 and its interacting RNA helicase Brr2, form part of the U4/U6.U5 tri-snRNP subunit of the spliceosome and, collectively, are mutated as a frequent cause of autosomal dominant retinitis pigmentosa (RP)[3–7]. Strikingly, RP-linked mutations

**Fig. 1 | Generation and characterisation of RPE and retinal organoids from RP13 and isogenic controls (RP13-Cas9). A** Schematic of iPSC-RPE differentiation; (**B**) Ezrin (green) and collagen IV (red) indicate that RP13-Cas9 iPSC-RPE have more protein present at the apical and basal membranes, respectively, relative to positions of nuclei (blue, Hoescht). Scale bar is 10 μm; (**C**) Functional characterisation of RP13 and RP13-Cas9 RPE cells. From left to right: Concentration of PEDF in apical transwell compartment of RP13 iPSC-RPE ($4123 \pm 1576$ ng/ml, $n = 3$) was significantly higher for RP13-Cas9 control ($2467 \pm 1472$ ng/ml, $n = 3$); concentration of VEGF in the basal compartment was similar between RP13 ($10.9 \pm 0.7$ ng/ml, $n = 3$) and RP13-Cas9 iPSC-RPE ($12.3 \pm 2.1$ ng/ml, $n = 3$); transepithelial resistance (TEER) values for RP13 (median = $222 \, \Omega \cdot cm^2$, LQ = $172 \, \Omega \cdot cm^2$, UQ = $298 \, \Omega \cdot cm^2$, min = $88 \, \Omega \cdot cm^2$, max = $510 \, \Omega \cdot cm^2$, $n = 102$) and RP13-Cas9-RPE (median = $227 \, \Omega \cdot cm^2$, LQ = $111 \, \Omega \cdot cm^2$, UQ = $292 \, \Omega \cdot cm^2$, min = $47 \, \Omega \cdot cm^2$, max = $525 \, \Omega \cdot cm^2$, $n = 118$) were not significantly different ($p = 0.12$); percentage of cells with internalised POS were not significantly different between RP13 ($58 \pm 13\%$, $n = 3$) and RP13-Cas9 ($54 \pm 17\%$, $n = 3$); (**D**) Schematic of iPSC-RO differentiation; (**E**) Bright field of ROs edge showing the brush border (scale bar 100 μm) and photoreceptors (Recoverin) comprised of red-green cones (OPN1LW/MW) and rods (Rhodopsin), and retinal ganglion cells (SNCG) cell (scale bar 50 μm); (**F**) Quantitative immunofluorescence analysis showing reduced cone, rod, and retinal ganglion cell presence in RP13 organoids (SNCG $p = 0.039$, OPN1 LW/MW $p = 0.016$, RHO $p = 0.040$, OPN1SW $p = 0.085$, PROX1 $p = 0.099$, AP2α $p = 0.41$). Data shown as mean ± SEM, $n = 6$; (**G**) Representative TEM images (left) showing elongated or swollen mitochondria, scale bar is 1 μm. Data shown in boxplots is median, box limits are $1^{st}$ (LQ) and $3^{rd}$ (UQ) quartiles, whiskers are maximum and minimum limits. Boxplots (right) quantifies reduced number of mitochondria in RP13 photoreceptor (PR) cell body (median = 70, LQ = 58, UQ = 95.3, min = 10, max = 150, $n = 3$) compared to RP13-Cas9 (median = 103.5, LQ = 77, UQ = 129, min = 33, max = 158, $n = 3$), $p = 0.0042$. Statistical significance was determined using paired 2-tailed $t$ test (**F** and **G**) or 2-tailed Student's $t$ test (**C**). **A** and **D** were created with BioRender.com.

are located in a C-terminal hotspot region of PRPF8 (encompassing the Jab1/MPN domain and its long "tail" extension) with a critical regulatory role on Brr2 helicase and ATPase activities required for the spliceosome activation[7]. Unlike other spliceosomal RNA helicases, after joining the spliceosome human Brr2 (hBrr2) remains stably associated, in complex with the PRPF8 Jab1/MPN domain, during the subsequent steps of the splicing cycle[2]. Clearly, hBrr2 helicase activity must be precisely regulated in the spliceosome. Previous in vitro biochemical studies with recombinant Brr2 and a PRPF8 C-terminal fragment in yeast have shown that this is mainly achieved by the PRPF8 Jab1/MPN domain acting as a stimulator/cofactor of Brr2 helicase and ATPase activities, whereas the PRPF8 "tail" extension inhibits Brr2 by blocking interaction with an RNA-binding "tunnel"[7–9]. However, the role of this key regulatory mechanism has not been elucidated in the human spliceosome and in human pre-mRNA splicing, so far. Fine-tuning of Brr2 activity by PRPF8 may play a role in splice site selection and maintaining optimal splice sites and splicing fidelity[9,10] and deciding alternative splicing events[11]. *PRPF8* RP patient mutations, through dysregulation of hBrr2 by PRPF8, therefore provide a valuable opportunity to delineate regulatory mechanisms of hBrr2 on human splicing in health and disease. In addition, PRPF-related RP mutations may also affect the assembly/stability of the tri-snRNPs, influencing the kinetics of B-complex formation. Nonetheless, it is unclear why these ubiquitous defects lead to a retinal-specific pathology in humans[12].

To reveal the role and importance of the fine-tuning of hBrr2 by PRPF8 in the human spliceosome and the consequences of its disruption by RP mutations, we herein generated patient iPSCs-derived retinal organoids (ROs), and retinal pigment epithelium (RPE) carrying the pathogenic *PRPF8* RP type 13 c.6926 A > C (p.H2309P) heterozygous missense mutation and their corrected isogenic controls (using the CRISPR/Cas9 genome editing). The p.H2309P mutation is located at the hinge that links the globular region of the Jab1/MPN domain to the tail, which is inserted into the hBrr2's RNA binding tunnel, and may de-stabilise the Jab1/MPN or the tail interaction with hBrr2. To compare the impact of this mutation between affected (retinal and RPE) and clinically unaffected tissues, we also generated kidney organoids (KiOs). Comprehensive image-based cellular and ultrastructural investigations of iPSCs-derived retinal tissues demonstrated that they recapitulate the typical RP disease phenotypes displaying multiple splicing, phenotypic and functional defects.

## Results

### Characterisation of RP13-RPE cells demonstrates impaired apical-basal polarity and decreased mitochondria incidence

We recruited three related RP type 13 patients (1A, 1B, 1C) and one with a de novo (2) *PRPF8* c.6926 A > C (p.H2309P) heterozygous pathogenic missense mutation, characterised by night blindness in childhood[13] (Supplementary data 1). We derived and characterised iPSCs reprogrammed from dermal skin fibroblasts (Fig. S1) and then generated isogenic normal controls for all patient iPSCs by correcting *PRPF8* c.6926A > C using CRISPR/Cas9 gene-editing (Fig. S2). Henceforward, we designate unedited iPSC lines with the c.6926A > C mutation as "RP13", whereas edited isogenic controls are "RP13-Cas9". We differentiated iPSCs from three RP13 and paired isogenic controls to RPE cells (Fig. 1A). RPE cell morphology and ZO-1 localisation were unchanged (Fig. S3A, B), but RP13-RPE epithelial cell polarity was impaired, indicated by disrupted expression of apical and basal markers (ezrin and collagen IV; Fig. 1B), and increased secretion of PEDF into the apical compartment (Fig. 1C). VEGF secretion into the basal compartment, transepithelial electrical resistance and the percentage of cells capable of phagocytosing photoreceptor outer segments (POS) (Fig. 1C) were unaffected in RP13-RPE. The median fluorescence intensity of iPSC-RPE cells was also unchanged between RP13 and RP13-Cas9 ($10482 \pm 2006$ and $12485 \pm 2567$ arb. units, respectively). Quantitative transmission electron microscopy (TEM) analysis of RPE ultrastructure (Fig. S3C–L) revealed a significant decrease in mitochondria in RP13 compared to RP13-Cas9 RPE (Fig. S3C), but no obvious abnormalities in microvilli or other subcellular organelles such as melanosomes (Fig. S3D–L).

### RP13-photoreceptors are characterised by increased degeneration and death, and decreased mitochondria incidence

RP13 and RP13-Cas9-iPSCs were differentiated into ROs for 210 days, displaying the presence of all retinal cell types (Fig. 1D, S4A). Both sets of ROs formed typical laminated neuroepithelia of a comparable thickness. In RP13-Cas9-ROs a dense "brush border" of photoreceptor outer segments was observed from day 180 of differentiation (Fig. 1E). Notably, this was missing in RP13-ROs, which also showed decreased immunostaining for cone (OPN1LW/MW), rod (rhodopsin) and retinal ganglion cell (SNCG) markers (Fig. 1E, F), suggesting an overall reduction in red/green cone and rod photoreceptor numbers, recapitulating the RP disease phenotype. TEM analysis indicated that the apical layer in RP13-ROs had frequent intercellular gaps (Fig. S4B), nuclei were frequently apoptotic and fragmented, suggesting photoreceptor cell loss as reported previously for RP11-ROs[14]. Notably the incidence of mitochondria was decreased in the photoreceptor cell body (Fig. 1G).

Single-cell RNA-Seq of all RP13 and RP13-Cas9-ROs was carried out on day 210 of differentiation. Following quality control, 63,875 cell transcriptomes were integrated and clustered based on highly expressed markers (Supplementary data 2) using Seurat (Fig. S5A). RP13-ROs had a higher proportion of degenerating rods, characterised by high *MALAT1* expression typical of retinal cell stress and degeneration[15] (Fig. S5B, C) consistent with TEM analysis (Fig. S4B). Analyses of differentially expressed genes (DEGs) (Fig. S5D, E) revealed

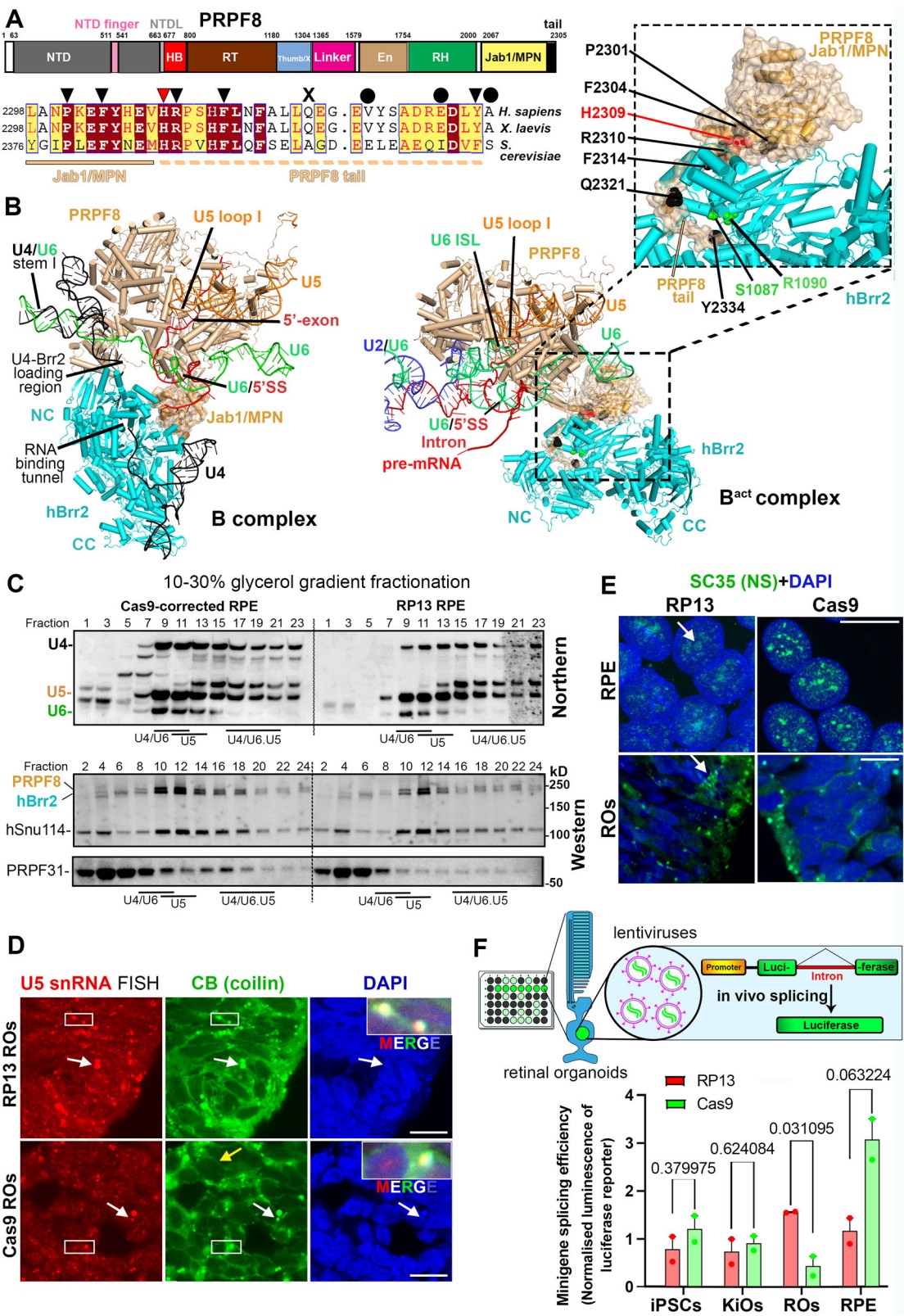

significant down-regulation of the cone (*ARR3*) and rod transcripts (*NRL*), connecting cilia (*RP1*) and phototransduction (*CNGB1, PDEA6A, GNAT1, GNGT1, GNB3*) in RP13-photoreceptors (Supplementary data 2). Retinal and photoreceptor degeneration were among the most affected "disease and function" subcategories (Fig. S5F). The phototransduction pathway was one of the most affected functional categories in both RP13-rod and cone photoreceptors (Fig. S5G).

**RP13-KiOs do not display obvious cellular or phenotypic defects**

To investigate whether the aberrant features of RP13-derived ROs might be either tissue-specific or more general, we differentiated iPSCs into KiOs that displayed nephrons and collecting duct structures (Fig. S6A–E). Localisation of markers for the apical cell surface (ezrin), basal cell surface (collagen IV) and tight junctions (ZO-1) were identical in RP13- and RP13-Cas9-KiOs (Fig. S6F). None of the RP13-KiOs

**Fig. 2 | The p.H2309P mutation in the PRPF8 Jab1/MPN tail does not disruptU4/U6.U5 tri-snRNP stability. A** Domain organisation of human PRPF8 (top), indicating RP mutations in PRPF8 Jab1/MPN and C-terminal tail regions (middle). NTD, N-terminal domain; NTDL, NTD linker; HB, helical bundle; RT, reverse transcriptase-like; En, endonuclease-like; RH, RNase H-like domain. Sequence alignment of the PRPF8 C-terminal region (bottom). Triangles, circles and cross indicate missense, frameshift and nonsense mutations, respectively; (**B**) 3D structures of PRPF8 (wheat), hBrr2 (cyan) and catalytic core RNAs in B and B^act spliceosomes (PDB IDs: 6AHD, 5Z56). Pre-mRNA, U6, U2 and U5 snRNAs shown in red, green, blue and orange, respectively. The hBrr2's catalytically active N-terminal (NC) and inactive C-terminal (CC) cassettes are labelled. The 5'SS/U6 ACAGA-box, pre-mRNA 5'-exon/U5-loop I, U4/U6 stem I, hBrr2 loading region on U4 snRNA and PRPF8 Jab1/MPN domain labelled as indicated on B-complex. Inset shows 3D structure of PRPF8 Jab1/MPN domain (wheat) complexed to hBrr2 (cyan) within spliceosomal B^act, showing PRPF8 C-terminal tail insertion into the hBrr2 RNA binding tunnel. p.H2309 and other RP-linked residues are highlighted by red or black spheres, respectively. RP-linked mutations (S1087L, R1090L) in hBrr2's RNA binding tunnel are indicated (green); (**C**) Glycerol gradient fractionation of RP13 or Cas9-RPE whole cell extracts analysed by Northern blotting (top) or Western blotting (bottom). Additional bands seen in the Northern blot of control fractions 1-7 that are under-represented in the RP13 fractions are due to nonspecific binding of the probes to the whole cell extracts nucleic acids; (**D**) RNA-FISH labelling of U5 snRNA (red) in Cajal bodies (anti-coilin, green; arrows indicate clusters) in RP13-Cas9 and RP13 photoreceptors. Insets show magnified selected regions. Scale bar 10 μm; (**E**) Immunostaining of RPE and ROs with SC35 showing dispersion of nuclear speckles in RP13-photoreceptor and RPE cells. Scale bar 10 μm. The experiments were repeated twice in **C** and **D** and three times in (**E**, **F**); Bar graphs showing splicing efficiency of the intervening intron. To control for viral transduction efficiency, the luminescence of intron-containing transcript was normalised against the intronless transcript ($n = 2$ biologically independent samples). Two-tailed $t$ test $p$-values shown above the bars. The schematic was created with BioRender.com.

displayed cyst formation (data available upon request), suggesting that cellular expression of pathogenic endophenotypes is confined to photoreceptors and RPE cells.

## The *PRPF8* p.H2309P mutation significantly affects splicing efficiency and causes the dispersion of nuclear splicing speckles in human retinal cells only

Previously, equivalent RP-linked mutations in the yeast PRP8 C-terminal Jab1/MPN domain and its extending tail (Fig. 2A) have been shown to impair tri-snRNP formation in yeast due to the disruption of Brr2 interaction with mutant PRP8 leading to a U5 maturation defect[7]. Moreover, in the cryo-electron microscopy structure of the human spliceosome complex B, hBrr2 is juxtaposed to the U4/U6 stem I and the single-stranded region of U4 is bound to the hBrr2's RNA-binding tunnel[16], and after the unwinding of stem I and release of U4 by hBrr2, its RNA-binding tunnel is occupied by the Jab1/MPN tail, in the B^act-complex[17,18] (Fig. 2B). Thus, we first assessed snRNP glycerol gradient fractionation profiles in iPSC-RPE cells. snRNAs and selected tri-snRNP proteins were then visualised by Northern and Western blotting, respectively (Fig. 2C). The tri-snRNP profile was unaffected, suggesting that the p.H2309 mutation does not change tri-snRNP stability (Fig. 2C, top panel). However, the level of U4 snRNA was reduced in RP13-RPE compared to the RP13-Cas9 control (Fig. S7E). The level of free hBrr2 was unchanged (Fig. 2C, bottom panel), and both PRPF8 and hBrr2 proteins complexed in stoichiometric amounts suggesting that the human tri-snRNP complex is stable under physiological conditions in RPE cells. Co-staining for PRPF8 and SC35 confirmed the localisation of PRPF8 in the nucleus within the splicing speckles in both RPE and ROs, indicating the incorporation of the PRPF8 mutant into the splicing machinery (Fig. S7A). We also assessed the morphology of nuclear speckles, compartments that store spliceosomal subunits, and the localisation of snRNAs and snRNPs into Cajal bodies, where snRNPs are assembled and matured (Fig. 2D, E and S7B). Although RNA-FISH and immunostaining for coilin showed normal localisation of U5 snRNA to Cajal bodies in RP13-RPE and photoreceptors, Cajal body compartments were larger in RP13-photoreceptors (Fig. 2D). Surprisingly, nuclear speckles in RP13-iPSCs and KiOs were structurally similar to controls, but both RP13-RPE and ROs had dispersed nuclear speckle clusters (Fig. 2E and S7C). Enlarged nuclear speckles have previously been reported in cells with disrupted splicing[19], suggesting that *PRPF8* p.H2309P mutation more severely affects splicing activity in RPE and ROs.

To assess endogenous splicing activity in different RP13 patient tissues, we transduced two lentiviruses expressing luciferase minigene transcripts with or without an intervening intron (Fig. 2F). These luciferase transcripts have been modified to have short mRNA and protein half-lives making them more sensitive to splicing changes[20]. Luciferase was down-regulated in RP13-RPE, indicative of lower splicing efficiency, but not in iPSCs or KiOs showing that *PRPF8* p.H2309P mutation has affected the RPE cells more severely. Surprisingly, RP13-ROs had significantly increased luciferase levels, indicative of enhanced splicing efficiency/kinetics due to the *PRPF8* p.H2309P mutation. This result is in agreement with previous data for RP mutations (S1087L and R1090L) in the RNA-binding "tunnel" of hBrr2, which are positioned in close proximity to the PRPF8 C-terminal tail, where they may also disturb PRPF8-mediated hBrr2 regulation[21] (Fig. 2B, inset). These hBrr2 RP mutations enhanced the splicing of minigene reporters, but this led to the increased usage of cryptic splice sites. Taken together, these results demonstrate that RPE and ROs are the most affected tissues for splicing efficiencies and morphological changes of splicing compartments upon *PRPF8* p.H2309P mutation.

## Disruption of PRPF8-mediated regulation of hBrr2 by the *PRPF8* p.H2309P mutation strongly leads to cryptic splice site selection and impairs alternative splicing in retinal tissues

To investigate in detail the importance of PRPF8 Jab1/MPN-mediated regulation of hBrr2 in human splicing and the consequence of its impairment by the p.H2309P mutation we next performed bulk RNA-Seq in iPSCs, KiOs, RPE and ROs derived from the four RP13 and RP13-Cas9-iPSCs. Substantially higher numbers of differentially expressed genes (DEG) were identified in ROs and RPE, compared to KiOs and iPSCs (Fig. 3A and Supplementary Data 3). Analysis of DEGs (Supplementary Data 4) revealed the most significantly down-regulated gene ontology (GO) functional categories in ROs were extracellular matrix and focal adhesions (Fig. 3B), and processes related to primary cilia (Supplementary Data 4). Up-regulated DEGs in ROs were enriched for RNA splicing (Fig. 3C). In RPE, down- or up-regulated DEGs were enriched for extracellular matrix organisation (Fig. 3B), or ribosomal transcripts (Fig. 3C) and nonsense-mediated decay (Supplementary Data 4), respectively. Extracellular matrix DEGs were also enriched in both iPSCs and KiOs (Fig. 3B, C).

We interrogated bulk RNA-Seq datasets for differential splicing using MAJIQ and rMATS. MAJIQ calculated the percent-spliced-in index (PSI) values and their standard deviation for each transcript (Fig. 3D). RP13-derived cells and retinal organoids had significantly increased splicing variation, suggesting alternative splicing deviation, but with a tissue-specific effect (RO $p = 5.20 \times 10^{-29}$; RPE $p = 1.98 \times 10^{-9}$; KiOs $p = 1.51 \times 10^{-3}$; iPSCs $p = 0.462$). Next, rMATS identified candidate alternative splicing events (ASEs) in annotated splice sites that were dysregulated between RP13-derived cells and organoids compared to RP13-Cas9 controls (Supplementary data 5, 6). Cryptic splicing events (CSEs) were defined as those using unannotated splice sites (Supplementary data 7). Greater numbers of differential ASEs as well as CSEs ($p_{adj} < 0.05$ and inclusion difference $> 5\%$) were identified in ROs and RPE, compared to KiOs and iPSCs (Fig. 3E and Supplementary data 5,

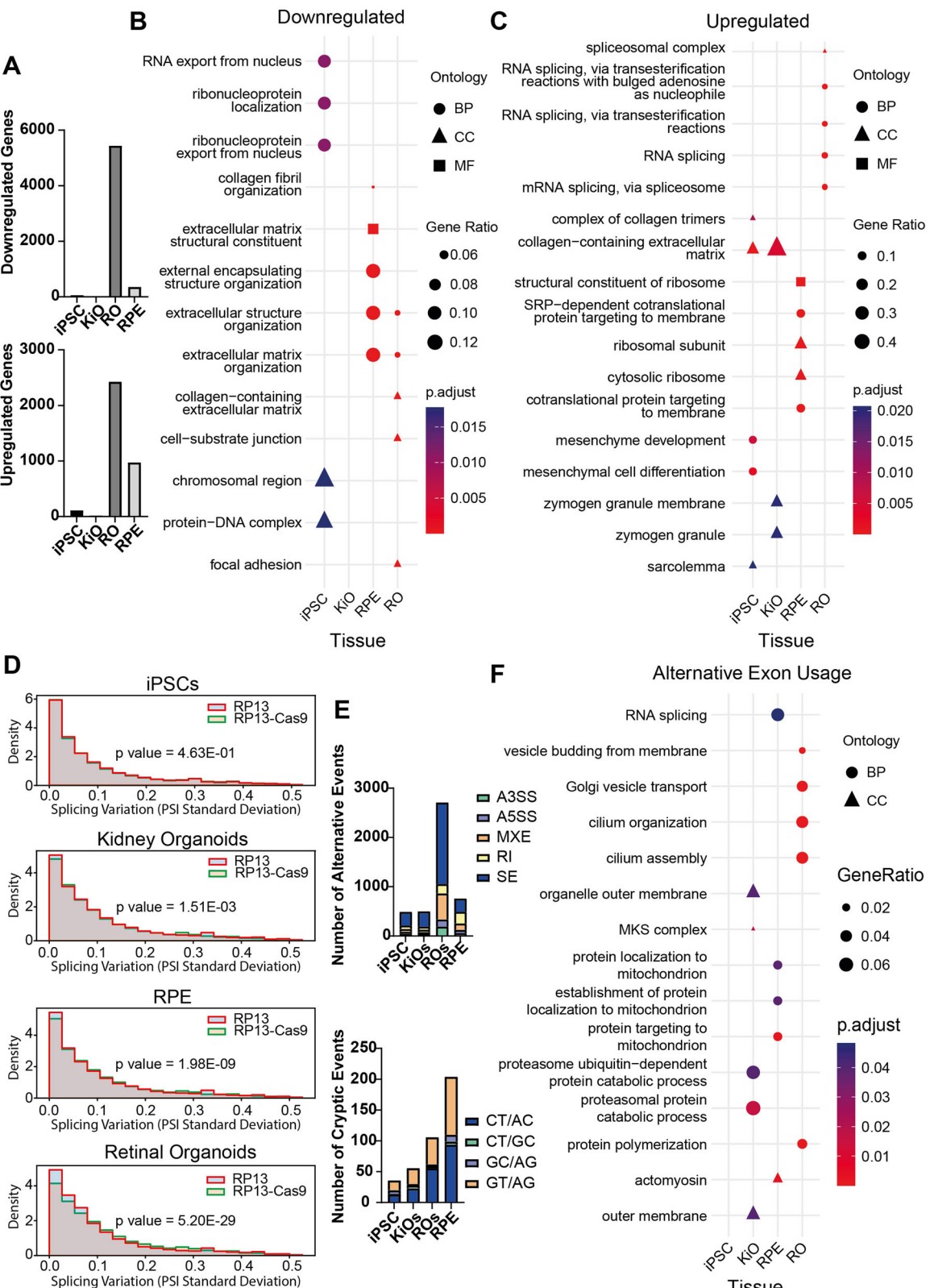

**Fig. 3 | Bulk RNA-Seq analysis of RP13- and RP13-Cas9-derived cells and organoids. A** Bar charts showing the higher number of DEGs in RP13-ROs and RPE cells; (**B**) GO enrichment analysis of genes downregulated and (**C**) upregulated in RP13-tissues, as determined using DESeq. Between 13 and 16 terms with the lowest adjusted *p*-values are displayed; (**D**) Density histogram showing the standard deviation of Percent Spliced In (PSI) values in RP13 and RP13-Cas9 derived cells and organoids, as measured using MAJIQ. Each data point was calculated using an f-test, but the overall p value in each graph between RP13 and RP13-Cas9 was obtained using an independent *t* test (**E**)

Bar charts showing the higher number of differential ASEs as well as CSEs in RP13-ROs and RPE cells; (**F**) GO enrichment analysis of genes identified by rMATS as exhibiting differential exon usage. Fifteen terms with the lowest adjusted *p*-values are displayed. Abbreviations: BP – biological process, CC- cellular component, MF – molecular function, SE – skipped exon, RI -retained intron, MXE – mutually exclusive exons, A3SS – alternative 3' splice site, A5SS – alternative 5' splice site. **B**, **C,** and **F**) One-sided Fisher's Exact Test with *p*-value adjustment for multiple comparisons (Benjamini & Hochberg) was carried out.

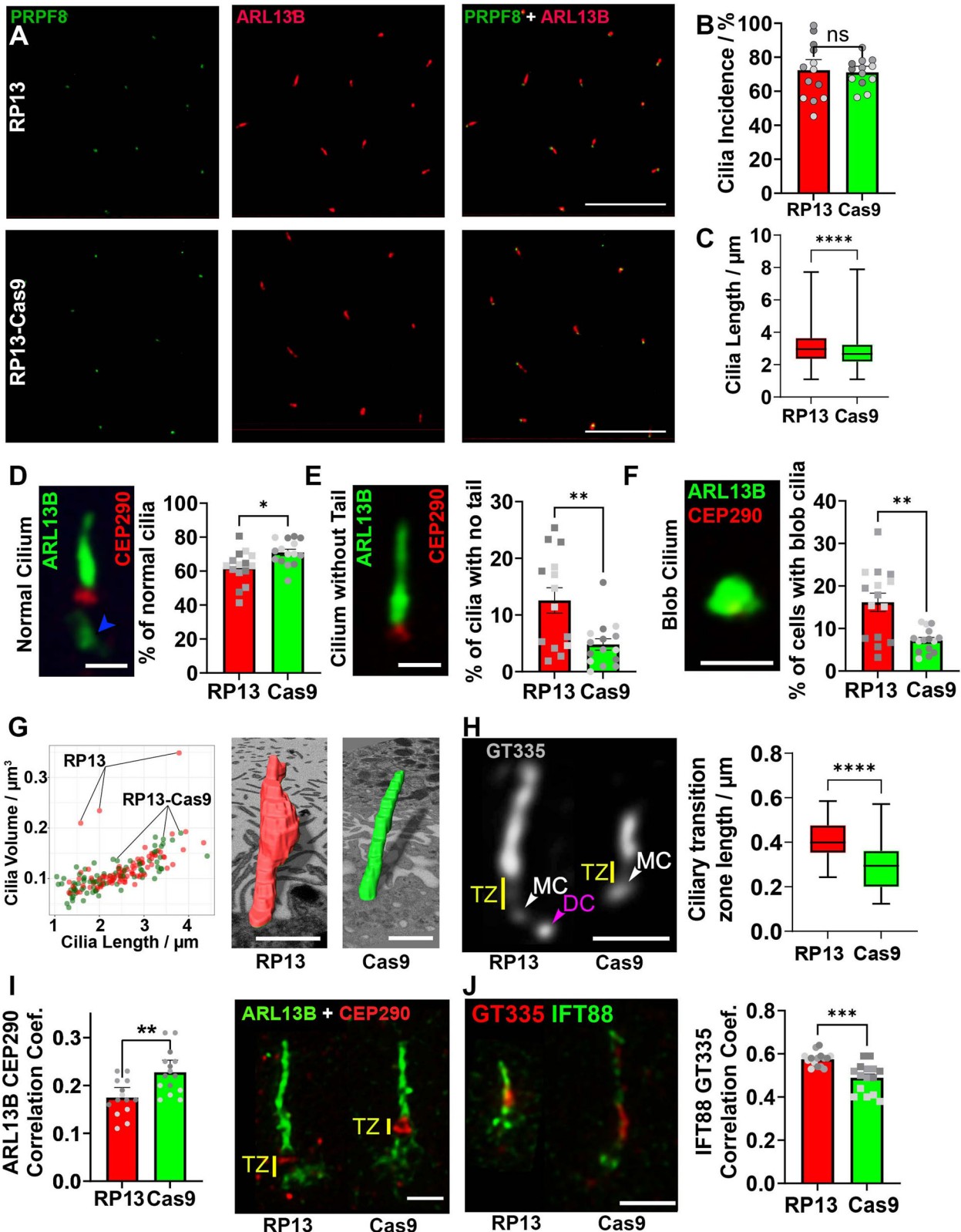

7). RPE had a higher fraction of retained introns compared to ROs and iPSCs, consistent with previous results[14], and in ROs, most ASEs were skipped exons (Fig. 3E and Supplementary data 5). GO enrichment analysis (Supplementary data 6) in ROs revealed that primary cilium assembly and cilium organisation were the most significantly enriched categories (Fig. 3F). The ciliary MKS complex and proteasome-mediated ubiquitin-dependent protein catabolic processes were enriched categories for KiOs, whereas RPE was enriched for mito-chondrial function and RNA splicing (Fig. 3F). There was no significant enrichment of any GO categories in iPSCs.

Altogether, these results demonstrated that disruption of PRPF8-mediated regulation of hBrr2 in the spliceosome by the p.H2309P mutation leads to profound changes in alternative splice sites selection in ROs as well as enhanced usage of cryptic/aberrant splice sites in RPE

**Fig. 4 | RP13- RPE cilia have increased length, abnormal morphology and altered transition zone structure. A** PRPF8 localisation to proximal ciliary membrane (ARL13B) in RP13 and RP13-Cas9-RPE cells, scale bar 10 µm; (**B**) Bar chart of cilia incidence, $n = 15$ fields of view from 3 donors and controls; (**C**) Boxplot of cilia length, $n = 3901$ and 3630 for RP13 (median = 2.96 µm, LQ = 2.36 µm, UQ = 3.63 µm, min = 1.09 µm, max = 7.72 µm, $n = 3901$) and RP13-Cas9 (median = 2.66 µm, LQ = 2.19 µm, UQ = 3.24 µm, min = 1.10 µm, max = 7.89 µm, $n = 3630$), $p = 2.4 \times 10^{-53}$, respectively where n is the total number of cilia from 3 donors and controls; (**D**) Representative image of a normal cilium, scale bar 1 µm. Blue arrowhead indicates "tail-like" morphology. Bar graph quantifies the percentage of normal cilia, $n = 1499$ and 1986 cilia analysed in Cas9 and RP13-RPE cells respectively ($p = 0.0099$); (**E**) Representative image of cilium without proximal ARL13B localisation, scale bar 1 µm. Bar graph quantifies the percentage of cilia without this localisation in RP13 and RP13-Cas9-RPE; $n = 15$ fields of view from 3 donors and controls ($p = 0.0033$); (**F**) Representative image and quantification of abnormal "blob"-shaped cilium, scale bar 1 µm; $n = 15$ fields of view from 3 donors ($p = 0.00049$); (**G**) Ciliary volume and length quantified using SBF-SEM, $n = 184$ individual cilia from 3 donors and controls, scale bar 1 µm; (**H**) iPSC-RPE cilia labelled for polyglutamylated tubulin (monoclonal antibody GT335), scale bar is 1 µm. The white arrowhead indicates the mother centriole (MC), and the purple arrowhead indicates the daughter centriole (DC, in RP13-RPE only). The yellow line indicates the ciliary transition zone (TZ). Boxplot quantifies average TZ length in RP13 (median = 0.40 µm, LQ = 0.35 µm, UQ = 0.46 µm, min = 0.24 µm, max = 0.59 µm, $n = 27$) and RP13-Cas9 (median = 0.29 µm, LQ = 0.21 µm, UQ = 0.36 µm, min = 0.12 µm, max = 0.57 µm, $n = 36$) iPSC-RPE cilia; n is number of cilia ($p = 1.25 \times 10^{-6}$); (**I**) ARL13B and CEP290 localisation, with bar graph expressing co-localisation as Pearson's correlation coefficient, $n = 15$ fields of view obtained from 3 donors and controls ($p = 0.0031$); scale bar 400 nm; (**J**) Representative GT335 and IFT88 localisation using super-resolution microscopy, scale bar 1 µm. Bar graph quantifies mean IFT88 fluorescence intensity, $n = 15$ fields of view obtained from 3 donors and controls ($p = 0.00012$). **B, C, E–J** results are mean ± SEM; statistical significance analysed by two-tailed Student's t-test with Welch's correction (*P-value 0.05, **P-value < 0.01, ***P-value < 0.001, ****P-value < 0.0001).

and ROs. These corroborates the critical function of hBrr2 RNA helicase in maintaining the optimal splice sites essential to maintain splicing fidelity.

## RP13-RPE cells display ciliary ultrastructural and organisation defects

To assess the possible effect of the *PRPF8* p.H2309P mutation on ciliogenesis and cilia function, as markers of cellular endophenotypes, we measured primary cilia length and cilia incidence using the ciliary membrane marker ARL13B. PRPF8 localised to the base of RPE cilia (Fig. 4A). In RP13 ROs, cilia length and incidence were significantly decreased (Fig. S8A), but both were unaffected in KiOs (Fig. S8B). RP13 RPEs had unaffected cilia incidence (Fig. 4B), but cilia were significantly longer (Fig. 4C) and many had abnormal morphologies (Fig. 4D–F). Abnormal cilium ultrastructure was also observed using SBF-SEM, with pronounced swelling of the ciliary membrane in 3.3% of RP13 RPE cilia (Fig. 4G). These structural defects were reflected by disrupted ciliary organisation in RP13-RPE, comprising increased separation between ciliary axoneme and basal body (Fig. 4H) and a defective ciliary transition zone assessed by colocalization of ARL13B and transition zone marker CEP290 (Fig. 4I). IFT88, a marker of ciliary trafficking (Fig. 4J), displayed significantly increased levels in RP13-RPE cilia compared to RP13-Cas9, consistent with the *PRPF8* mutation disrupting normal ciliary trafficking.

## The *PRPF8* p.H2309P mutation causes accumulation of active spliceosomes and poly(A) + RNA in unique co-transcriptional splicing clusters in human photoreceptors

To further investigate splicing abnormalities in retinal cells, we determined levels of the activated spliceosome by Western blotting for phosphorylated SF3B1 (p-T313-SF3B1), a marker of activated spliceosomes (B^act)[22], and the U4/U6 protein PRPF31 that is phosphorylated in the pre-activated spliceosome (complex B) (Fig. 5A). Phosphorylated SF3B1 (p-SF3B1) was 2.5-fold increased and phosphorylated PRPF31 was marginally increased in RP13-RPE compared to RP13-Cas9 controls (Fig. 5B). In RP13-Cas9 photoreceptors, p-SF3B1 immunostaining showed that active spliceosomes preferentially localised at the periphery of nuclei (Fig. 5C). Moreover, p-SF3B1-marked spliceosomes colocalised with SC35-positive nuclear speckles, with a distinct polarity relative to DAPI-stained DNA foci, and significantly accumulated in RP13 photoreceptors (Fig. 5C inset and 5D, E) consistent with defects in splicing kinetics. To confirm co-transcriptional splicing at these splicing clusters in photoreceptor cells, we immunostained ROs for RNA polymerase II (RNAPII), a marker for active transcription sites. RNAPII localised within and around nuclear speckle clusters, but also at the periphery of the nuclear chromatin (Fig. 5H), consistent with a previous study in mouse retina[23]. Poly(A)+ RNAs localised to splicing clusters, as expected, but at significantly increased levels in the splicing clusters of RP13-photoreceptors (Fig. 5F, G), suggesting accumulation of aberrant mRNAs. Therefore, these results demonstrate the presence of a unique splicing compartment at the nuclear periphery of human photoreceptor cells consisting of sub-compartments Cajal bodies, nuclear speckles and active transcription sites and spliceosomes (Fig. 5I). Moreover, these corroborate our RNA-Seq study indicating accumulation of transcripts with differential alternative splicing or cryptic splicing in the RP13 patient photoreceptors.

## The *PRPF8* p.H2390P mutation alters PRPF8 crosslinking pattern with U6 snRNA and disrupts binding to snoRNAs

To delineate the effect of p.H2390P mutation on the interactions of PRPF8 with RNAs, particularly at the catalytic centre of the spliceosome, we performed individual nucleotide-resolution crosslinking and immunoprecipitation (iCLIP-seq) for PRPF8 in RP13-iPSC derived tissues. The most frequent RNA interactors of PRPF8 across all tissues were U5, U6, and U4 snRNAs and the long non-coding RNA *MALAT1* (Fig. 6A and Supplementary data 8). We then assessed the PRPF8-snRNA crosslinking pattern (Fig. 6B) to determine whether p.H2390P mutation affected PRPF8 interactions at the spliceosome catalytic core[24]. PRPF8 and U5 snRNA interactions were largely unchanged (Fig. 6B, yellow highlights), corroborating our gradient fractionation results (Fig. 2C). Notably, the mutant PRPF8 displayed a significant and tissue-specific increase in the proportion of total hits over nucleotides 25–47 of U6 in RP13 retinal cells (RPE and ROs), but not iPSCs and KiOs (Fig. 6B, light green). This region of U6 contains the ACAGA box, an evolutionarily conserved sequence that base pairs with the 5'-splice site (5'SS) in the B-complex, and remains stable in the activated B^act, B* and C spliceosome complexes (Figs. 6C, 2B and S7D). Moreover, the formation of the U6/5'SS duplex triggers an extensive remodelling of the spliceosome that includes the relocation of hBrr2 and its loading onto U4 snRNA leading to the formation of the pre-activated spliceosome complex B (Fig. 2B). Consequently, hBrr2 is poised for U4/U6 unwinding leading to spliceosome activation. The interaction of PRPF8 with the U6 internal stem-loop (ISL, dark green) forms after the release of U4 by hBrr2 in B^act; however, this was unaffected by the *PRPF8* p.H2309P mutation (Fig. 6B, dark green), suggesting a sequence-specific alteration in PRPF8 interaction with U6 (Fig. 6D). Analysis of deletions in U6 sequencing reads demonstrated nucleotides 64–65 in the U6-ISL as the actual crosslinked nucleotides (average 4.3% of the reads carrying a deletion in this position across all samples). Inspection of the structure of the B^act complex shows that these nucleotides are adjacent to the PRPF8 N-terminal domain finger (Fig. S7D) making it competent for RNA-protein crosslinking. PRPF8 also showed an increase in the number of total hits on U4 over nucleotides 36–96 in RP13 retinal cells. Notably, this region of U4 contains the hBrr2 loading

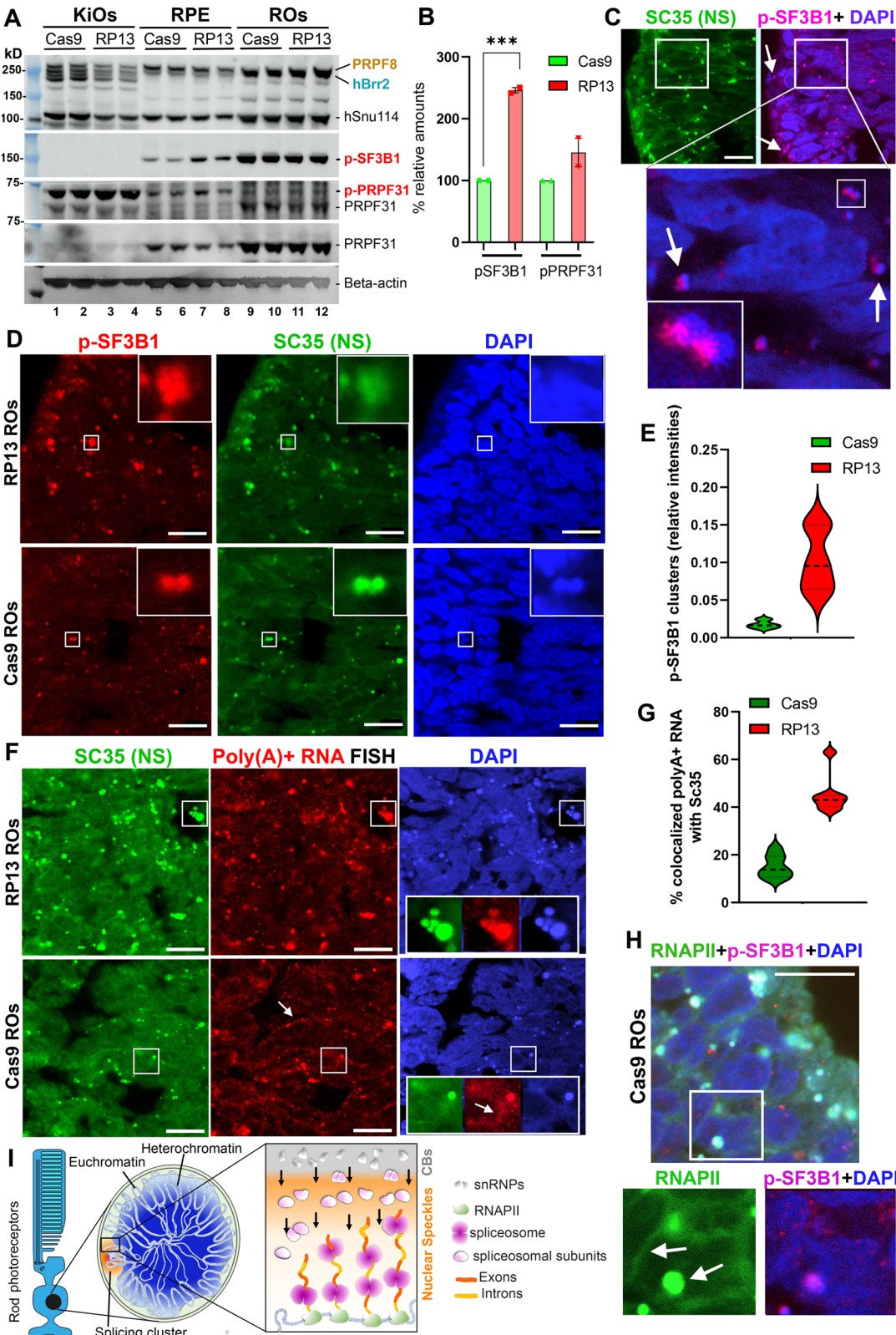

sequence, bound to hBrr2 in the B-complex. However, we did not detect deletions in U4 sequencing reads suggesting that the U4 binding might be indirect via other proteins (e.g., hBrr2) or U4 base pairing with U6, which is not disrupted in the iCLIP experiments. The *PRPF8* p.H2309P mutation also increased tissue-specific binding of small nucleolar RNAs (snoRNA) in retinal cells (RPE and ROs) but caused decreased binding in KiOs (Fig. S9A, B).

**The *PRPF8* p.H2309P mutation affects 5′-splice site recognition by PRPF8 of transcripts encoding ciliary proteins**

In coding regions, PRPF8 iCLIP tags were located predominantly in exon-intron or intron-exon boundaries. We used maximum entropy ("maxent") scores to assess the strength of the bound region as a potential 5′ (maxent5) or 3′ (maxent3) splice site[25]. Maxent3 scores were distributed normally (Fig. 7A), indicating no enrichment of 3′-

**Fig. 5 | Accumulation of active spliceosomes and poly(A)+ RNA in unique splicing clusters of human RP13 photoreceptors. A** Western blotting of the whole extracts from the KiOs, RPE and ROs tissues for the indicated splicing proteins. The beta-actin was used as a loading control for quantification; (**B**) Graphs showing quantification of p-SF3B1 (245.9 ± 4.4) and p-PRPF31 (145.4 ± 22.9) Western blot bands in RP13 RPE relative to the Cas9 controls (100.0 ± 0.0). The differences between RP13 and controls (two samples in each group) were only significant for p-SF3B1 (*P*-value = 0.0009), (****P*-value < 0.001, two-tailed *t* test); (**C**) In photoreceptor cells active spliceosomes marked with p-SF3B1 (red) are localised in SC35 (green) clusters adjacent to DAPI-stained DNA foci (blue) at the periphery of the nucleus. The insets show the magnification of the selected regions. The experiment was repeated three times, independently. Scale bar 10 μm; (**D**) Accumulation of active spliceosomes (p-SF3B1, red) in splicing clusters (SC35, green) of RP13 photoreceptors. The insets show the magnification of the selected regions. The experiment was repeated three times, independently. Scale bar 10 μm; (**E**) Quantification of the intensities of p-SF3B1 clusters in RP13 versus Cas9 control photoreceptors (*n* ≥ 120 cells analysed) in (**D**); (**F**) Increased levels of poly(A)+ RNA (red) within the splicing clusters (SC35, green) in RP13-photoreceptors, monitored by RNA FISH followed by immunostaining for SC35. The insets show the zoom of the selected regions. The arrow indicates RNAs localised to the nuclear periphery in the control. The experiment was repeated two times, independently. Scale bar 10 μm; (**G**) Quantification of the co-localisation of poly(A)+ RNA with SC35-positive clusters in RP13 versus RP13-Cas9-photoreceptors (*n* ≥ 150 cells analysed) in (**F**); (**H**) Immunofluorescence imaging of photoreceptors labelled with RNAPII (green), p-SF3B1 (red) and counterstained with DAPI (blue) showing concentration of RNAPII in the splicing clusters in photoreceptors. The inset shows the magnification of the selected region. The arrow indicates RNAPII localised to the nuclear periphery. The experiment was repeated three times, independently. Scale bar 10 μm; (**I**) Schematic representation of the nuclear architecture in human photoreceptors showing their unique chromatin organisation and splicing clusters. The schematic was created with BioRender.com.

splice sites in RNAs bound to PRPF8. However, maxent5 scores were distributed with a lower kurtosis in RP13 retinal cells (ROs and RPE), compared to iPSCs and KiOs (Fig. 7B, S9C), indicating a retinal tissue-specific effect. Weak 5′ (maxent5 < 3) and strong 5′SSs (maxent5 > 8) were preferentially crosslinked to PRPF8. We therefore identified individual transcripts with significant changes in binding (log₂ fold-change > 1 or < − 1; Fig. 7B) and performed enrichment analysis for weak and strong 5′SSs. RP13-ROs had significantly fewer weak 5′SSs bound to PRPF8 in transcripts encoding proteins mediating transcription, and ciliary functions (Fig. 7B, C), corroborating the enrichment of ciliary transcripts that undergo alternative exon usage in this tissue (Fig. 3F). Interestingly, the exons frequently bound by PRPF8 in RP13-ROs were characterised by thymidine-rich sequences, whereas the downregulated exons were cytidine-rich (Fig. 7E). This was not the case for the PRPF8 bound exons in RP13-RPE cells, KiOs or iPSCs (Figure S9F). AT-rich exons are known to localise to the nuclear periphery, recognised by exon definition and their alternative splicing leads to exon skipping[26]. RP13-RPE had slightly more weak 5′SSs bound to PRPF8 (Fig. 7B and Supplementary data 8) affecting similar pathways (Fig. 7D; RNA binding, nucleolus, nuclear lumen; cilium assembly, cilium). Weak 5′SSs appeared to be unaffected in both iPSCs and KiOs (Fig. 7B) but, nevertheless, significantly enriched GO pathways were for ciliary processes in both cell-types (Fig. S9D, E and Supplementary data 9). For KiOs, this corroborates the findings for alternative exon usage in ciliary transcripts of the MKS complex (Fig. 3F). We observed similar results for strong 5′SSs (Fig. 7B), with fewer bound to PRPF8 in RP13-ROs enriched in transcripts for transcription and signalling by Hedgehog, a ciliary-mediated signalling process (Fig. 7B, C). Notably, strong 5′SSs for transcripts encoding ciliary proteins were identified in RP13-RPE cells, consistent with the ciliary abnormalities observed in RPE (Fig. 4). However, there were also fewer strong 5′SSs in the KiOs and iPSCs (Fig. 7B) despite the lower number of alternative and cryptic splice events (Fig. 3E). This suggests that binding of PRPF8 to the strong 5′SSs is unlikely to affect splicing.

These results suggested that PRPF8-mediated regulation of hBrr2 plays a key role for the selection of 5′SSs by the spliceosome, a function previously reported for hBrr2[21]. Dysregulation of this mechanism by a *PRPF*-RP13 mutation predominantly affects splicing at weak (suboptimal) 5′SSs in ROs leading to mis-splicing of ciliary transcripts and the RP pathogenesis.

**TMT-based quantitative proteomic analysis demonstrates that ROs are the most affected tissue in RP13**

To confirm our transcriptomic data at the protein level, we next performed 10-plex tandem mass tag (TMT)-based quantitative mass spectrometry to identify proteomic differences between RP13-ROs, KiOs and RPE cells and their respective RP13-Cas9 controls. This identified proteins with differential expression (DE) that had 40% change in RP13 cells relative to controls (Supplementary data 10) and demonstrated more differentially expressed proteins (DEPs) in ROs (203) and RPE cells (54) compared to KiOs (4) (Fig. 8A–C). PRPF8 protein expression was unaffected, consistent with the expression of both wild-type and mutant transcripts detected by qRT-PCR in all tissues studied (Fig. S10A) and results from Western blotting (Fig. S10B). Consistently with the transcriptomic results, levels of most splicing factors were slightly increased in RP13-ROs, except for CWC15, a component of the PRP19 complex, which was significantly decreased (Fig. 8A). DE proteins in ROs were enriched for GO components and pathways including collagen-containing extracellular matrix, focal adhesion, spliceosome, endoplasmic reticulum, and lysosome (Fig. 8D and Supplementary data 10). For RPE, GO terms were vesicle membrane, endosome, phagosome, and retinol metabolism (Fig. 8E and Supplementary data 10). Integrated analysis of transcriptome and proteome changes revealed 1400 matched transcripts/proteins in ROs (Fig. 8F) and moderate correlation between transcript and protein changes (R = 0.26). In RPE, 386 matched transcripts/proteins had very low correlation, consistent with the conclusion that transcript levels are insufficient to predict protein changes (Fig. 8G).

Changes in the expression of several proteins previously involved in RP was noted in RP13-ROs. For example, CRYAB[27] is one of the most down-regulated genes in rod photoreceptors that was strongly affected at both transcript (Fig. S5E) and protein levels (Fig. 8A, F and Supplementary data 2). However, other proteins implicated in RP (e.g., SFRP2[28] were significantly upregulated in RP13-ROs (Fig. 8A). In RPE, several proteins associated with retinal degeneration were down-regulated (e.g., TIMP3, RDH5, PRELP, Fig. 8B). Our results demonstrate that ROs exhibit the most disturbed proteome due to *PRPF8* mutation and suggest that changes in the expression of proteins associated with retinal degeneration could cause PRPF8 RP.

## Discussion

RP is one of the most common forms of sight loss with a prevalence of about 1 in 4000 births and more than 1 million people affected worldwide[29]. Autosomal dominant inheritance accounts for ~30–40% of RP, with ~11% caused by mutations in *PRPFs*. Given the ubiquitous expression of PRPFs, it remains paradoxical as to why mutations in tri-snRNP components manifest as retinal-specific degeneration. The highly conserved scaffolding protein of the spliceosome PRPF8 and its interacting dual-cassette RNA helicase Brr2 form a large portion of the U4/U6.U5 tri-snRNP and are mutated in RP[3-7]. Moreover, the intriguing regulatory mechanism of PRPF8 on Brr2 helicase activity through the PRPF8 Jab1/MPN domain and its extending tail, where the RP type 13 (RP13) causative mutations accumulate, with unclear role in the spliceosome function and splicing, highlights PRPF8 as an important candidate to help us gain insight into spliceosome function and RP pathomechanism. Therefore, we reprogrammed fibroblasts with

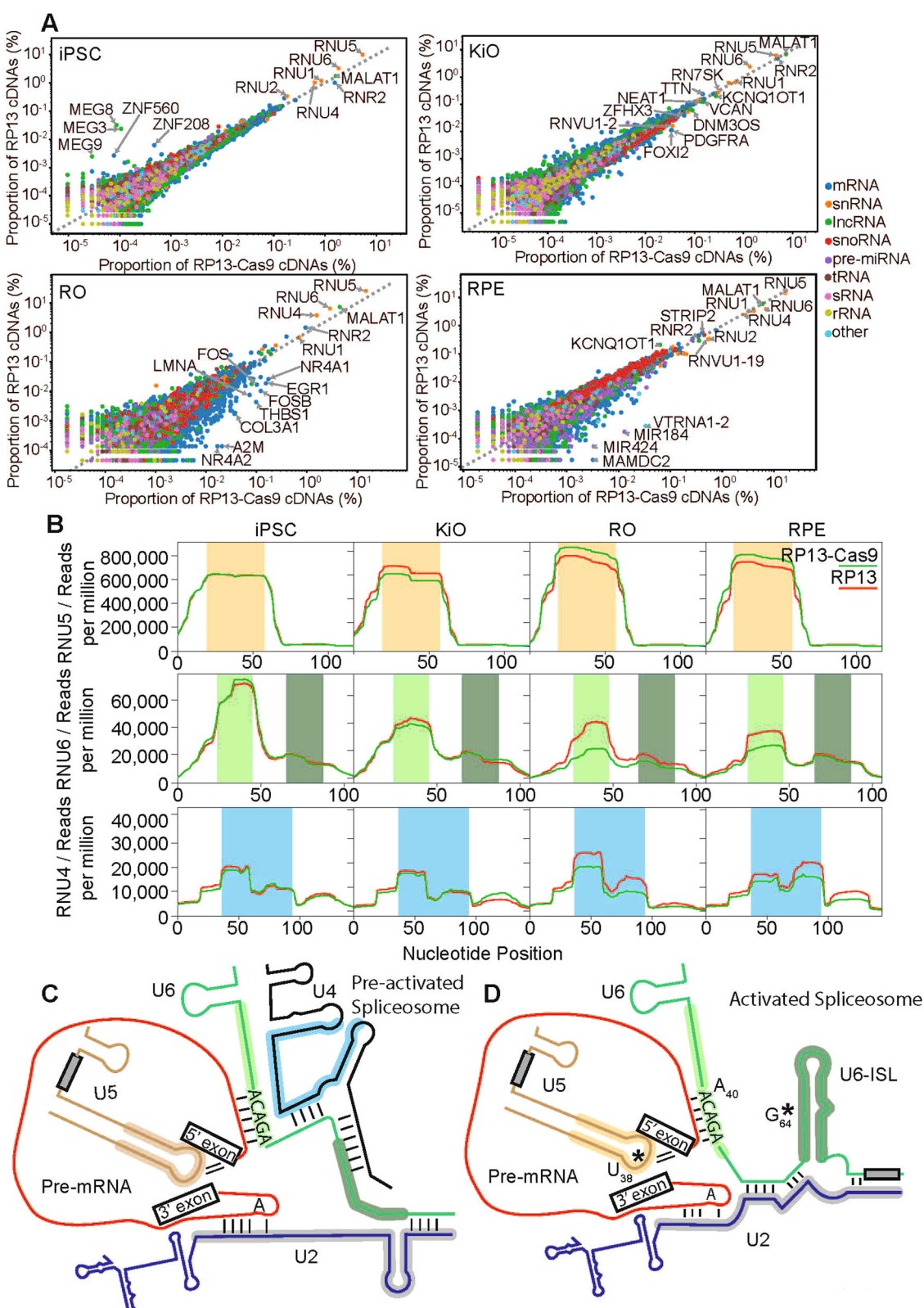

heterozygous *PRPF8* c.6926 A > C (p.H2309P) mutations into iPSCs. After correcting these iPSCs into isogenic controls by the CRISPR/Cas9 technology, we differentiated panels of matched iPSC-derived KiOs, ROs and RPE cell cultures. Here, we report the systematic integration of cellular and biochemical analyses, transcriptomic, PRPF8 inter-actome and proteomic datasets to reveal the importance of the fine-tuning of hBrr2 by PRPF8 in the spliceosome for pre-mRNA splicing and the functional consequences of its disruption by RP-related mutations, unravelling the mechanism of PRPF-RP pathogenesis. Our data indicate the critical regulatory role of PRPF8/hBrr2 for 5'-splice site recognition, the disruption of which affected numerous tissue-specific cellular endophenotypes. These comprised changes in the PRPF8 interaction pattern with the spliceosomal U6 snRNA, altered splicing efficiency and spliceosome kinetics, and organisation of

**Fig. 6 | PRPF8 iCLIP provides information on the interaction between PRPF8 and cognate snRNAs. A** Scatter plots showing the abundance of RNAs bound to PRPF8 in RP13- and RP13-Cas9 iPSC-derived tissues. Across all tissues, there is a relatively high level of U5 (*RNU5*), U6 (*RNU6*), and U4 (*RNU4*) snRNAs bound to PRPF8; (**B**) Line graphs showing the interaction profile between PRPF8 and U5 (*RNU5*), U6 (*RNU6*), and U4 (*RNU4*) snRNAs. A high number of reads suggests a strong interaction between PRPF8 and nucleotides at that position of the snRNA. Regions with strong binding to PRPF8 are highlighted in yellow (U5), light and dark green (U6), and blue (U4). Note the tissue-specific effect on PRPF8-U6 interactions in ROs and RPE, but not iPSCs and KiOs, comprising the binding of a 20-nucleotide region of U6 (nucleotides 26–46). The number of reads detected in RP13 tissues is plotted as a red line whereas those in RP13-Cas9 are plotted as green. An increase in binding between PRPF8 and a region of U6 (*RNU6*) (light green) was observed in both replicates of ROs and RPE cells. A similar increase in binding is apparent between PRPF8 and U4 (*RNU4*), although this effect was not reproducible across experiments; (**C**) Illustration showing the RNA-RNA interactions present in the pre-activated spliceosome (**B**). Most regions of the snRNAs identified as having a strong interaction with PRPF8 in (**B**) are highlighted in the colours used in (**B**), except for the dark green region in U6. The highly bound region of U6 that is highlighted in light green contains the ACAGA box; (**D**) Illustration showing the RNA-RNA interactions present in the activated spliceosome (B^act). All regions of the snRNAs identified as having a strong interaction with PRPF8 are highlighted in the colours used in (**B**). The dark green region of U6 maps to the U6-ISL which forms during B-complex activation.

nuclear speckles. These caused significant differential alternative splicing (e.g., ciliary transcripts) and cryptic splicing as well as retina-specific accumulation of aberrant poly(A)+ mRNAs and loss of photoreceptors.

The RP13-RPE cellular endophenotype includes a loss of apico-basal polarity, whereas RP13- ROs were characterised by degenerating photoreceptors which led to an overall reduction in photoreceptor numbers, mimicking the patient's phenotype in the advanced stages of RP. Transcripts in RP13-RPE undergoing differential exon usage, were significantly enriched in those encoding mitochondrial and splicing proteins, whereas ROs were enriched for ciliary function encoding proteins. RP13-RPE also had primary ciliary length and morphology defects. This corroborates findings of mis-splicing of ciliary transcripts and primary ciliary structural defects for RP11-RPE in our previous report[14]. Tissue-specificity of these defects suggest they are a major contributor to RP pathogenesis in RPE and photoreceptors. This also highlights the utility of iPSC-based models for mimicking patient phenotypes in the advanced stages of human RP, because the murine RP type 13 model carrying a heterozygous *Prpf8* p.H2309P mutation only manifests an RPE phenotype[30,31].

We observed retinal cell-specific alterations in splicing efficiency and disrupted spliceosome kinetics, assessing the proportion of splicing complexes by using markers for the B complex (phosphorylated PRPF31) and the activated spliceosome complexes (phosphorylated SF3B1). This revealed increased levels of active spliceosomes and accumulation of poly(A)+ RNAs, specifically in the nuclear speckles of retinal cells. Underlining the importance of tissue specificity, spliceosome kinetics were only marginally affected in HeLa cells modelling *SNRNP200* mutations causative for RP[21]. Interactome studies using iCLIP-Seq confirm that increased occupancy of PRPF8 at the U6 snRNA ACAGA-box occurred in a retinal tissue-specific manner. This region forms the U6/5′SS duplex inducing a series of events leading to hBrr2 relocation and its loading onto the U4 snRNA, and spliceosome activation. For pre-mRNA transcripts, PRPF8 preferentially bound to the 5′SS of pre-mRNA transcripts, consistent with previous work[32], but weak 5′SS were more affected by the *PRPF8* p.H2309P mutation in retinal cells. Interestingly, changes in PRPF8 binding to the affected exons with weak 5′SS were sequence-dependent (i.e., downregulated exons were cytidine-rich and upregulated exons were thymidine-rich). The 5′SS is initially defined by its base pairing with the U1 snRNA and after the release from U1, it pairs with the U6 ACAGA-box. Several proteins including PRPF8 help stabilise and maintain the U6/5′SS into the catalytic core at later stages of splicing. The direct role of PRPF8 in 5′SS recognition and splicing fidelity, which is more critical for genes with weak/suboptimal 5′SS[33], and its impaired function upon RP mutations in the C-terminal tail corroborates the regulatory role of PRPF8/hBrr2 within the spliceosome in splice site selection[7,34]. Several spliceosomal helicases/NTPases have been indicated in proofreading of splice site choice in yeast, including Prp5, Prp28, Prp2, Prp16, Prp22 and Prp43. Notably, the role of PRP16 and PRP22 helicases, respectively, in selection of alternative branch-point and 3′ splice sites have been documented[11]. In comparison with these helicases that are transiently recruited to the spliceosome, Brr2 is a permanent resident after its joining as part of the tri-snRNP. It contains two helicase cassettes, of which the N-terminal cassette is catalytically active, and several cis- and trans-regulatory mechanisms have been reported to fine-tune its activity in the spliceosome[3,9,35,36]. Previously, two RP-linked mutations (p.S1087L and p.R1090L) in the hBrr2's RNA-binding tunnel, in close proximity to the PRPF8 C-terminal tail that may weaken the interaction with the tail, were shown to result in a shorter interaction time of spliceosomes with pre-mRNA (resulting in higher $k_{off}$ rates) than the wild-type controls in FRAP experiments[21]. Higher splicing efficiency and increased 5′-cryptic splice-site selection of a reporter gene, suggested a role for hBrr2 in maintaining 5′-splice site identity. A genetic screen in *Caenorhabditis elegans* has identified a strong suppressor of 5′SS mutation and cryptic splicing in the unstructured N-terminal region of Brr2[10]. Moreover, truncation of this region in yeast has resulted in the reduction of U6 and U5, and accumulation of U1 in the B^act spliceosome[37]. Thus, a timely loading of hBrr2 on U4 and unwinding of U4/U6, upon formation of the 5′SS-ACAGA-box helix, is important for maintaining 5′SS in the catalytic centre of the spliceosome. It is conceivable that premature activation of hBrr2 or its impaired inhibition after U4 release may lead to the interaction of ACAGA-box with a second alternative or a cryptic 5′SS, or formation and accumulation of defective B^act, respectively. The key regulatory mechanism on Brr2 is induced by the PRPF8's Jab1/MPN domain and its extending "tail" that can either activate or inhibit Brr2 activity. This study demonstrates the importance of this mechanism in the human spliceosome, in vivo, for the selection of weak/ suboptimal 5′SSs, showing that its disruption in the RP13 disease state impairs alternative splicing and enhances cryptic splicing events, predominantly in retinal cells.

Differentially PRPF8-bound transcripts were enriched for those encoding ciliary components in retinal cells, corroborating our alternative exon usage data, and reiterating the importance of ciliary disruption in tissue-specific RP pathogenesis. In addition, we found increased binding of PRPF8 to C/D and H/ACA box snoRNAs in RP13 retinal cells. Apart from mediating post-transcriptional modification of small non-coding RNAs (such as ribosomal RNAs and snRNAs), snoRNAs have been implicated in alternative splicing and micro-RNAs production[38]. Moreover, a fraction of snoRNAs have been found associated with spliceosomes[38]. Although the role of snoRNAs in oncogenesis[39] and autoimmune diseases have been documented, their function in hereditary retinal diseases is unclear and remains to be explored. These data suggest that they may play a direct or indirect role in the aetiopathogenesis of PRPF8 RP.

Our proteomics data confirmed that only ROs had protein expression that correlated with transcript levels, with significant enrichment of genes/proteins previously implicated in photoreceptor degeneration and RP. The functional consequences of alternative splicing on protein expression and isoform diversity remain unclear, with older studies identifying a single major isoform for most genes with little variation due to alternative splicing[40–42]. A more recent integrative approach demonstrated correlation between levels of

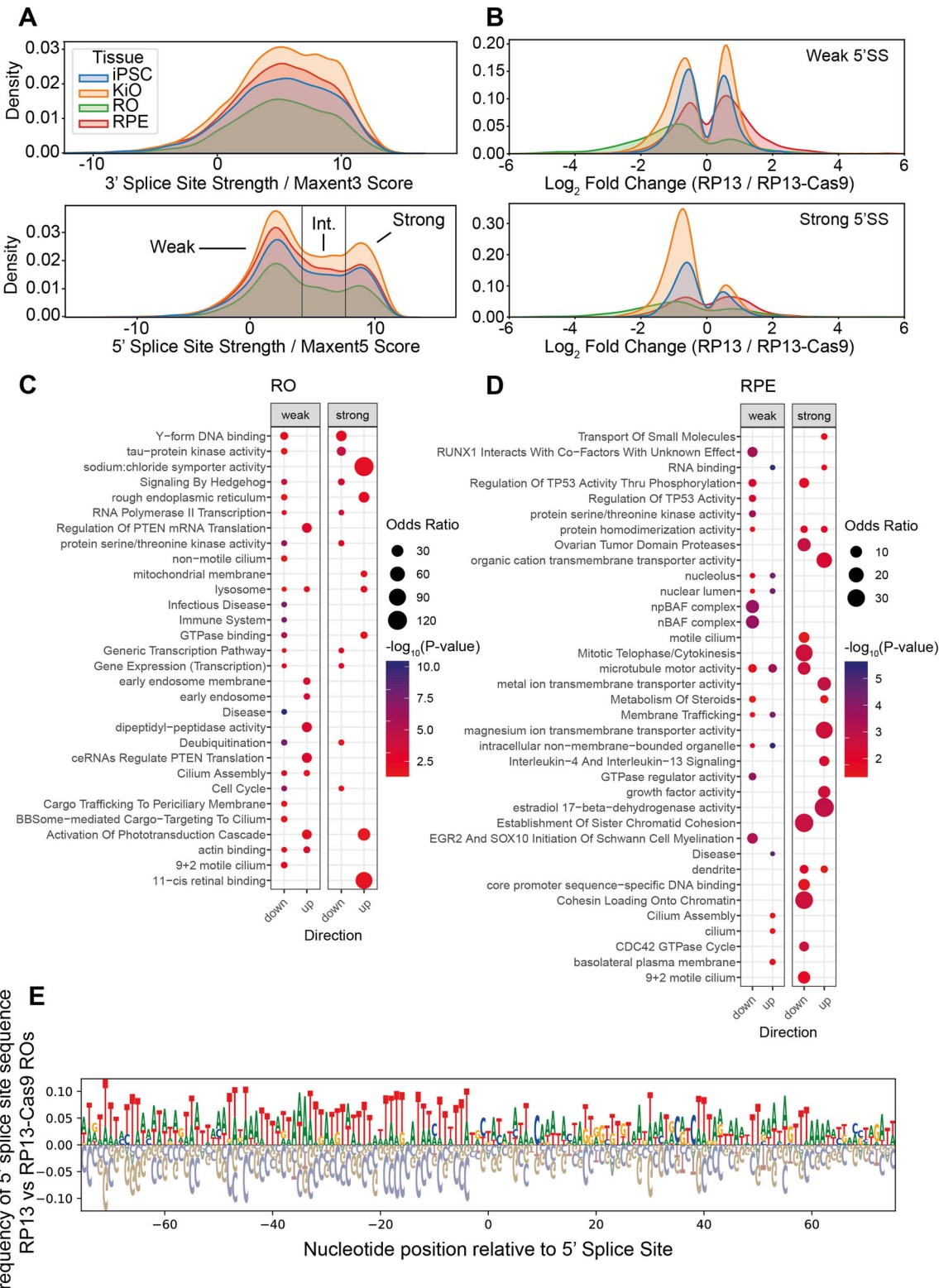

**Fig. 7 | PRPF8 iCLIP provides information on the interaction between PRPF8 and pre-mRNA. A** PRPF8 does not selectively bind to 3' splice sites. The 3'SS strengths of bound pre-mRNA fragments are normally distributed, as would be expected for random sequences. PRPF8 selectively binds to 5'SSs. The biphasic distribution of 5'SS strengths has two maxima (approximate maxent5 scores of 2 and 9) separated by a local minima. These were categorised into three 5'SS strengths: weak (maxent5 score <3), strong (maxent5 score > 8), and intermediate (maxent5 score between 3 and 8); (**B**) Histograms showing fold changes (RP13/RP13-Cas9) of weak and strong 5'SSs. KiOs and iPSCs histograms have more pronounced peaks with low fold change values. RPE and ROs histograms have flatter distributions with a greater proportion of the values present in the tails of the curve. The kurtosis of the ROs and RPE graphs is lower than iPSCs and KiOs (Fig. S10C); (**C**) Gene set enrichment analysis of differentially bound weak and strong 5'SSs in ROs showing enrichment for transcripts encoding proteins mediating transcription and ciliary functions (e.g., RNA polymerase II transcription, cilium assembly, cargo trafficking to periciliary membrane); (**D**) Gene set enrichment analysis of differentially bound weak and strong 5'SSs in RPE. Direction down and up refers to positive and negative fold changes in (RP13/RP13-Cas9), respectively. GSEAPY interface for Enrichr with Fisher's exact test was used in (**C**, **D**); (**E**) Gene sequence plot showing differential inclusion of nucleotides at weak 5'SSs with upregulated or downregulated binding to PRPF8 in ROs (**B**).

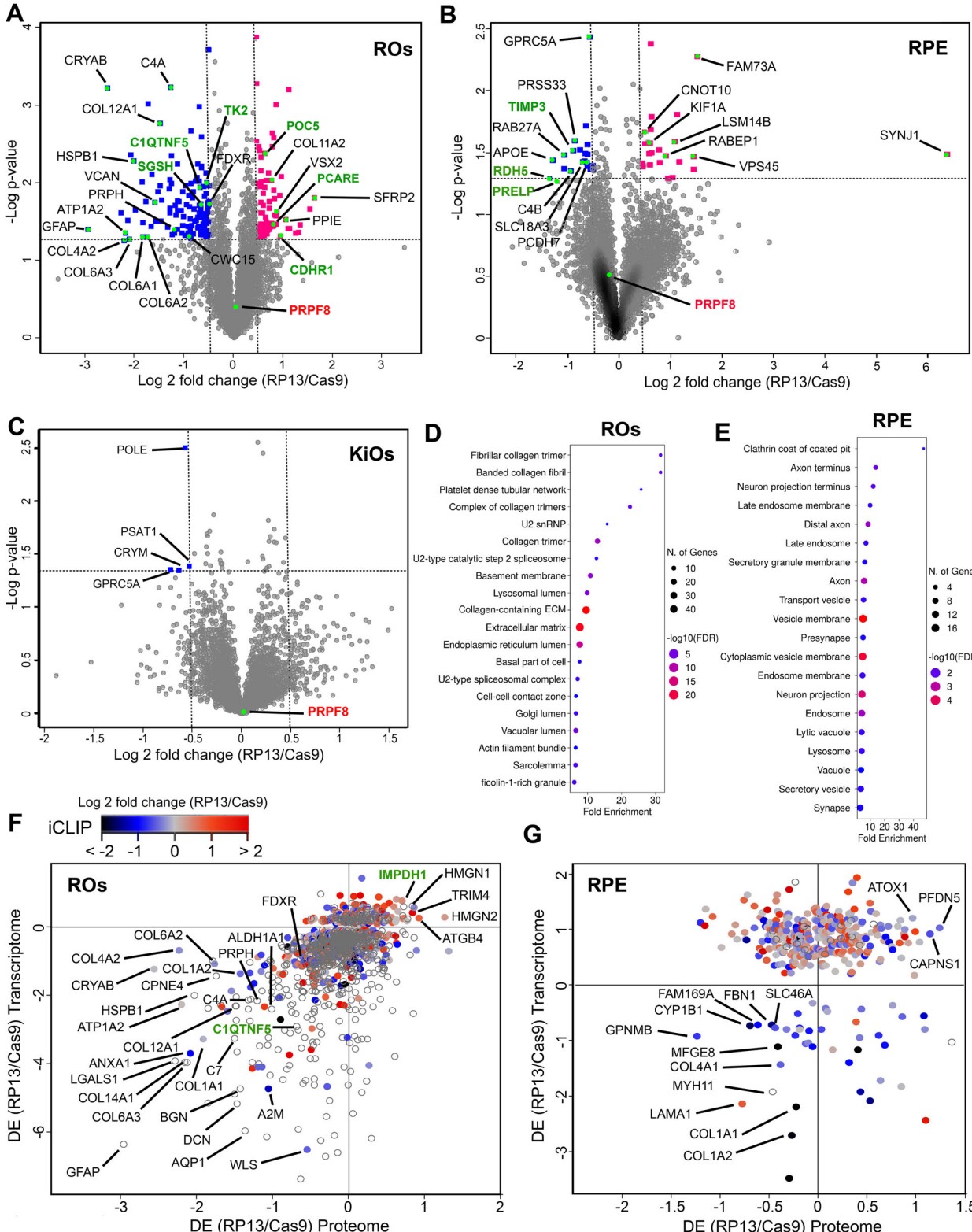

**Fig. 8 | Differential expression of proteome between RP13 and RP13-Cas9. A–C** Volcano plots showing the log2 fold change against the two-tailed t-test derived −log10 statistical *p*-value for all proteins (represented by 2 or more unique peptides) and differentially expressed (DE) between RP13- and RP13-Cas9-derived tissues (**A**) 4535 unique proteins, DE = 203), RPE (**B**) 4574 unique proteins, DE = 56) and KiOs (**C**) 4294 unique proteins, DE = 4). Different RP13- and RP13-Cas9-ROs had consistent high correlation coefficients of *R* = 0.96–0.99, respectively. Upregulated and downregulated proteins are shown in magenta and blue, respectively. PRPF8 is labelled in red and proteins associated with retinal degeneration are labelled in green; (**D, E**) Top GO terms (in the domain of cellular components) related to proteins with significant differential expression in ROs (**D**) and RPE (**E**); (**F, G**) Correlation between proteomics, transcriptomics and iCLIP data in RP13 versus control ROs (**F**) and RPE (**G**); Scatter plot of protein abundance ratios against corresponding mRNA ratios is plotted. The changes in iCLIP ratios are displayed using a colour gradient. Selected hits are labelled. The colourless datapoints represent proteins that were not identified in the iCLIP experiments.

transcripts perturbed by constitutive or alternative splicing and levels of the encoded proteins[43]. We observed this correlation only in RP13-ROs, suggesting that the functional interpretation of transcript diversity, in the form of encoded protein isoforms, is a process limited to neurosensory retinal cells. This may comprise a unique vulnerability in the molecular pathogenesis of RP, and other inherited retinal conditions since other cell-types appear to have redundancy in maintaining their proteome and are unaffected by mis-splicing.

We also observed that human photoreceptors have a unique nuclear architecture, in comparison to other cell-types, consistent with a previous report describing them specifically in mouse rod photoreceptors[23,44]. We observed localisation and clustering of active transcription and splicing machineries adjacent to DAPI-stained DNA foci at the nuclear periphery, with the clusters enriched for markers of Cajal bodies (coilin) and nuclear speckles (SC35) as well as RNAPII. This suggests that both spliceosomal subunit assembly and co-transcriptional splicing could occur in these clusters, ensuring high levels of splicing efficiency. In general, splicing efficiency is thought to be determined by sequence features, transcription level and concentration of splicing machinery. Recent work has demonstrated that transcripts localised near the nuclear speckles exhibit higher splicing efficiency due to a higher concentration of spliceosomes[45,46]. We speculate that the special organisation of splicing clusters in human photoreceptor cells may drive higher splicing efficiency in these cells, consistent with the characteristics of the retina including larger amounts of snRNAs and spliced housekeeping genes[47] as well as a higher number of alternative splicing events compared to other tissues. However, this architecture results in a distinct 3D genome organisation in photoreceptors that may comprise a further vulnerability in the molecular pathogenesis of RP and explain the origin of retinal-specific disease phenotypes. It could also be the cause of higher splicing of the reporter gene observed specifically in RP13 retinal organoids. There were no changes in PRPF8 protein localisation in RP13 mutant retinal cells compared to isogenic RP13-Cas9 controls, or in U4/U6.U5 tri-snRNP stability, suggesting that the mutant PRPF8 isoform could compete with the wild-type protein in a dominant negative manner. Therefore, gene augmentation approaches are unlikely to be therapeutically effective for RP patients with PRPF8 mutations. More complex strategies that target the mutant allele (*e.g.*, prime editing), or gene augmentation combined with down-regulation of the mutant transcript, should be pursued in developing effective treatments for RP13-RP.

## Methods

The research complies with all relevant ethical regulations and was performed according to the protocols approved by Yorkshire and the Humber Research Ethics Committee (REC ref: 15/YH/0365). The study was conducted in accordance to the criteria set by the Declaration of Helsinki.

### Human subjects

Informed consent for research studies, according to the protocols approved by Yorkshire and the Humber Research Ethics Committee (REC ref: 15/YH/0365), was obtained for all study participants. Information on the patient and control participants is provided in Supplementary Data 1. The study design and conduct complied with all relevant regulations regarding the use of human study participants.

Detailed descriptions of all other methods are available in Supplementary Methods available in the Supplementary Information. Key reagents and software are provided in Supplementary Table 1, included in the Supplementary Information.

### Statistical analysis

Statistical analysis was performed using Prism (GraphPad, USA). Data were tested for normality using the Anderson-Darling test.

Comparisons between variables and statistical significance between groups were performed using ANOVA, the two-tailed Student's *t*-test or paired *t*-tests. Data were plotted as mean values with error bars representing standard error of the mean (SEM) unless indicated otherwise. Additional information such as N values (the number of independent biological replicates) is presented in figure legends. Statistical significance of pair-wise comparisons is indicated by asterisks: $*P < 0.05$, $**P < 0.01$, $***P < 0.001$, and $****P < 0.0001$. Significance was defined as a *p*-value $< 0.05$.

### Reporting summary

Further information on research design is available in the Nature Portfolio Reporting Summary linked to this article.

## Data availability

The trimmed FASTQ data for bulk RNA-Seq of all samples included in this study were uploaded to SRA under the accession number BioProject ID PRJNA989762 and GEO under the accession number GSE236702. The mass spectrometry proteomics data have been deposited to the ProteomeXchange under the accession number PXD043645. The single cell RNA-Seq data have been deposited to the GEO under the accession number GSE235866. The iCLIP-Seq data have been uploaded to Annotare under the accession number E-MTAB-13171. A source data file has been provided with the manuscript. Source data are provided in this paper.

## Code availability

No new codes were used in this study.

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

## Acknowledgements

The authors are grateful for financial support from Medical Research Council (MR/T017503/1) to M.L., S.M.J., C.A.J., S.M.N.G. and R.R.A. Biotechnology and Biological Sciences Research Council (BB/T004460/1) to M.L., J.C. and R.Q. Retina UK (GR601) to M.L., R.R.A., S.M.J., S.N.G. and H.U. and the Peter & Sue Cheney bequest to the University of Leeds (CAJ). H.U. was supported by a grant from the Deutsche Forschungsgemeinschaft (SFB1286). S.M.J. was supported by a grant from NIMAD (989278). Data generated using the Gatan 3View system and Hitachi TEM was made possible by financial support from BBSRC grant BB/M012093/1 and BB/R013942/1 to T.D. We would like to thank Newcastle University flow cytometry, genomics and microscopy core facilities, and the facility for light microscopy at the Max-Planck Institute for Multidisciplinary Sciences. Special thanks to Carolina Lascelles and Morag Raynor at the NGS Facility and Leeds 'Omics at the University of Leeds, and the Durham University computational biology facility for help with technical assays and RNA-Seq analyses. We are grateful to Ciaran Morrison for providing the CEP290 antibody, Nicolas Sergent, Zeiss Research Microscopy, for help with AiryScan2 imaging, the Durham University computational biology facility for help with technical assays and RNA-Seq analyses and Monika Raabe and Ralf Pflanz for assistance in LC-MS.

## Author contributions

R.A., M.G., K.S. and C.Y. data acquisition and analysis, contributed to figure preparation and manuscript writing. A.L. and E.J.R.V. data analysis, contributed to figure preparation and manuscript writing. Y.J., R.L., M.K. and M.M.M. data acquisition and analysis. T.D., J.C., B.D., F.G., C.S., A.W., M.K. and L.A. data acquisition. A.S., C.S.B., M.K.A. and R.Q. data analysis. C.I., M.M. and D.H.S. sample collection. D.J.E. data acquisition

and fundraising. R.L. and H.U. data acquisition, analyses, and fundraising. R.R.A., C.A.J., S.M.J. and S.N.G. study design, data analysis, fundraising and manuscript writing. M.L. study design, data analyses, fundraising, manuscript writing and overall co-ordination of this study. R.A., M.G., C.Y., K.S., A.L. and E.R.V. contributed equally to this work. All authors approved the final version of the manuscript.

## Competing interests

The authors declare no competing interests. The funders had no role in study design, data collection and analysis, decision to publish, or preparation of the manuscript.

## Additional information

[1]Biosciences Institute, Newcastle University, Newcastle, UK. [2]Leeds Institute of Medical Research, University of Leeds, Leeds, UK. [3]Department of Biosciences, Durham University, Durham, UK. [4]Leeds Omics, University of Leeds, Leeds, UK. [5]Max-Planck-Institute for Multidisciplinary Sciences, Göttingen, Germany. [6]Institute of Clinical Chemistry, University Medical Center Göttingen, Göttingen, Germany. [7]Newcells Biotech, Newcastle, UK. [8]Electron Microscopy Research Services, Newcastle University, Newcastle, UK. [9]Department of Informatics, University of Bergen, Bergen, Norway. [10]Göttingen Center for Molecular Biosciences, Georg August University of Göttingen, Göttingen, Germany. [11]Centre for Cell and Gene Therapy, Kings College London, London, UK. [12]Department of Medical Genetics and Medical Genetics Research Center, School of Medicine, Mashhad University of Medical Sciences, Mashhad, Iran. [13]These authors contributed equally: Robert Atkinson, Maria Georgiou, Chunbo Yang, Katarzyna Szymanska, Albert Lahat, Elton J. R. Vasconcelos. [14]These authors jointly supervised this work: Robin R. Ali, Sushma-Nagaraja Grellscheid, Colin A. Johnson, Sina Mozaffari-Jovin, Majlinda Lako. ✉e-mail: c.johnson@leeds.ac.uk; sina.mozaffari-jovin@mpinat.mpg.de; majlinda.lako@ncl.ac.uk

