## [Peer Review File · Nature Communications]

PRPF8-mediated dysregulation of hBrr2 helicase disrupts human spliceosome kinetics and 5'-splice-site selection causing tissue-specific defectsReviewers' Comments:

Reviewer #1:

Remarks to the Author:

Atkinson and colleagues are proposing a manuscript where they delved into the mysteries of a spliceosomal scaffolding protein PRPF8 in which mutations are causative for an inherited retinal degenerative disease, retinitis pigmentosa-type 13. Using a comprehensive combination of RP13 patient-derived iPSCs, and isogenic controls where the mutation had been corrected using CRISPR/Cas9, as well as RPE cells, retina organoids and kidney organoids, they were able to decipher the mechanistic underpinning of RP13 and why PRPF8 mutations cause retina-confined pathology, despite PRPF8 being abundantly expressed in our bodies. I found the manuscript pleasurable to read and the topic interesting. The work can have substantial clinical relevance with respect to therapeutic approaches. I did not find serious flaws in the manuscript, but I have a few major and several minor comments that should be addressed before publication.

Major:

- A typical clinical RP manifests as photoreceptor dysfunction and death whereas the inner retina remains relatively intact, including the retinal ganglion cells (RGCs). Still, the authors show significant loss of the RGC marker SNCG (Fig. 1F) in their RP13 retinal organoids (RO). On the other hand, bipolar, horizontal and amacrine cell markers show similar expression in RP13 and control ROs. Isn't it controversial that RGCs would be so clearly affected in a RP phenotype, and does it affect the translational validity of the model if this is the case?
- The authors should use the maximum granularity in data presentations. The authors use bar graphs with individual data points, or scatter plot, in e.g. Fig. 4B, but not throughout the manuscript. Using scatter plot, the reader can easily see the sample size and make more informed conclusions themselves. Along these lines, the difference in e.g. Fig. 1G does not look $P < 0.01$ significant to me. Similarly, the effect in 4H does not look $P < 0.0001$.
- The materials & methods part is lengthy (+6000 words), which is understandable because so many techniques have been used. Still, I suggest trying to cut it a bit shorter by e.g. referencing to earlier work. Nat Commun guide to authors say: "Methods should be written as concisely as possible and typically do not exceed 3,000 words but may be longer if necessary. We encourage you to deposit any step-by-step protocols used in your study in Protocol Exchange, an open resource maintained by NPG."
- Also, some of the figure legends exceed the suggested <350 words, particularly in Fig. 2. Guide to authors also suggest abstract length of approximately 150 words. Current abstract is ~200.

Minor:

- In the abstract, "rod degeneration" and "photoreceptor loss" are essentially synonyms. Can remove "rod degeneration".
- There are several flaws in the references list. For example, the authors cite preprints of old papers, although proper bibliography must exist. Citation 7 has two different years? Citation 20's doi is to BioRxiv reprint and not to their Nat Commun paper. Etc. etc. Please correct throughout.
- Before citing Figure S5F, I suggest to edit: "Retinal and photoreceptor degeneration were among the most affected "disease and function" categories".
- Before citing Fig. 2D, cannot say "significantly larger" if there is no quantification of the phenomenon. Please edit text.
- In page 6, I suggest removing the word "Strikingly". Having seen/read the earlier results, the amount of DEGs was not that striking to me at least.
- In page 10, I am not sure if the current citation 26 is the best. In that paper, the correlation of transcriptomic and proteomic data actually looks relatively strong. I don't see that literature citation here is all that necessary.
- In early discussion, please do a fact check. I think Hartong and Dryja state RP prevalence at 1:4000. Today, many references state 1:3000-1:4000. Also, according to Hartong and Dryja, PRPF mutations may not account for 15-20 % of adRP, but a lower amount at ~ 11 %.
- Please soften the statement: "...collectively unravelling the mechanisms of RP pathogenesis". It

remains complex and there are so many types of RP...

- Figure 5A. The authors should show full Western blots as supplementary data.

Reviewer #2:

Remarks to the Author:

Heterozygous mutations that affect certain splicing factors lead to forms of retinitis pigmentosa (RP) in humans. Presently, it is not understood how the resulting defects in ubiquitously required splicing factors lead to a tissue-specific disease phenotype. Atkinson et al. addressed this question with respect to a RP13-linked mutation in the large spliceosomal scaffold, PRPF8. Specifically, they investigated a mutation that leads to a H2309P exchange in PRPF8. The affected residue is positioned at the base of a Jab1/MPN domain at the C-terminus of PRPF8. This domain has been shown to bind the spliceosomal helicase BRR2 and regulate BRR2 helicase activity, in part via a C-terminal tail that can be inserted into the BRR2 RNA-binding tunnel. To investigate effects of H2309P-PRPF8, the authors reprogrammed fibroblasts with the relevant heterozygous mutation derived from four patients into iPSCs, and differentiated those into retinal pigment epithelium (RPE) cells, retinal organoids (ROs), and, for comparison, kidney organoids (KIOs). They also generated isogenic controls, in which the mutation was corrected by CRISPR/Cas9-based genome engineering. They used image-based cellular and ultrastructural investigations to demonstrate that iPSC-derived retinal tissue exhibits RP disease phenotypes. They then conducted transcriptomics and quantitative proteomics studies, revealing splicing defects as well as abnormal cilia and functional defects in retinal cells and organoids. Splicing defects were confirmed via a lentiviral splicing reporter. The authors also characterized structurally unique splicing clusters in human photoreceptor cells at the nuclear periphery. Using iCLIP-seq, the authors showed that H2309P-PRPF8 exhibited altered binding to 5'-splice sites of transcripts that encode ciliary proteins as well as altered interactions with a key functional region of U6 snRNA and snoRNAs in retinal organoids.

The authors developed a powerful model system to study the tissue-specific effects of a RP-linked mutation affecting a key splicing factor. The battery of powerful state-of-the-art cellular biological, multi-omics and biochemical approaches they employed to characterize disease phenotypes on the tissue, cellular and molecular level is truly impressive. While this reviewer is no expert in the techniques applied, the work appears to be technically sound and the results support the conclusions drawn. This work not only uncovers disease principles of a particular mutation affecting a splicing factor, but also provides a blueprint for delineating how mutations that affect ubiquitously required factors manifest in a tissue-specific manner. Of particular interest is the discovery of a special organization of subnuclear splicing compartments in human photoreceptors. The manuscript is well-written and should be accessible to a wide audience. This reviewer has only very few specific comments/questions:

1. Can the authors discuss in more detail how dysregulation of BRR2 by PRPF8 RP13 variants may affect 5'-splice site selection, as they detect in their data? Specifically, how PRPF8-mediated regulation of BRR2 might affect the selection of suboptimal 5'-splice sites?
2. The authors describe that the H2309P exchange may "destabilize" the PRPF8 C-terminal tail that can intermittently inhibit BRR2, suggesting that H2309P-PRPF8 may inhibit BRR2 less efficiently. Previously, it was shown that RP13-linked variants of the PRPF8 Jab1/MPN domain can have diverse effects on BRR2 helicase activity (RNA Biol 11, 298; 2014). Is it known how a H2309P variant of PRPF8 (or of the Jab1/MPN domain) affects BRR2 activity? Can this effect explain the splicing defects observed?
3. Do the authors attribute any disease relevance to the altered snoRNA interactions detected for H2309P-PRPF8?

Reviewer #3:

Remarks to the Author:

Atkinson et al. investigated the impact of pathogenic RP mutations in the PRPF8 gene on splicing in humans. They utilized iPSCs-derived retinal organoids (RO), retinal pigment epithelium (RPE) and clinically unaffected kidney organoids (KiOs) from patients carrying PRPF8 RP type 13 c.6926A>C (p.H2309P) heterozygous missense mutation, and their isogenic controls generated by correcting the mutation using CRISPR/Cas techniques. They then performed a comprehensive set of analyses, including imaging, RNA-cross-linking, transcriptomic and proteomic assessment to investigate differences in splicing control between these cells. Their results reveal impairment of alternative splicing predominantly in ciliary and retinal-specific transcripts.

PRPF8 plays a pivotal role in the spliceosome function, orchestrating the catalytic core of the spliceosome through interactions with pre-mRNA, U2, U5, U6 snRNAs, and various protein components. In yeast, the C-terminus of Prp8 is known to regulate the ATPase and RNA helicase activities of Brr2, which are essential for spliceosome activation. This specific region is susceptible to RP mutations that affect the interaction between Prp8 and Brr2. However, the mechanism responsible for the tissue-specific effects of these mutations remains unknown. In this study, the authors employed a range of analytical methods to investigate the impact of RP mutations on the splicing process in humans. They demonstrate that RP mutations can affect PRPF8-U6 interaction, the splicing efficiency, splicing kinetics, and the organization of nuclear speckles. Specifically, they have a tissue-specific impact on alternative splicing in 5' splice sites selection. While the precise mechanism driving the tissue specificity of these mutations remains unresolved, it represents a significant advancement in our efforts to understand how these mutations give rise to RP phenotypes.

This work involves an extensive analysis of genome-wide data, but there are instances where the interpretations of data lack clarity. To make this work more accessible to broader readers, it would be beneficial for the authors to provide clear explanations. Moreover, the finding that RP mutations impair 5' splice site recognition is in line with the current notion that the 5' splice site is freed from U1 binding for interaction with U6 before the action of Brr2. It may be worthwhile to incorporate this concept into the discussion. Specific comments addressing inaccuracies in the use of certain terminology and in the description of the outlined mechanisms are detailed below.

1. p.3, line 7, "Fine-tuning of Brr2 activity may modulate splice site proofreading...". Using the term "splice site selection" is more suitable than "proofreading". There is no evidence for splice site proofreading in any known human cases. The prevalence of alternative splicing, which accommodates a wide array of splice site sequences, adds complexity to the concept of proofreading.
2. Fig. 2C: It is not apparent to me that the amount of U4 is reduced in RPE as the intensities of U5 and U6 decreased correspondingly. This may be attributed to the possibility of lighter exposure in the RPE blot compared to the control. Furthermore, several bands observed in fractions 1-5 of the control that are absent in RPE sample, which need further explanation or discussion.
3. Figure 2F shows that the PRPF8 p.H2309P mutation had a minor impact on splicing efficiency in iPSC and KiO cells, while it had a more pronounced effect on RPE and RO cells, albeit in opposing directions. The authors should provide an explanation for this divergence between RPE and RO cells. It is conceivable that distinct introns may be regulated differently, given that only one specific intron was examined in this experiment. To draw a conclusion, the authors should replicate this experiment with other introns.
4. Fig. 3: The abbreviations used, such as BP, CC, MF in 3BCF, and A3SS, A5SS, MXE ... etc. in 3E, should be defined in the figure legends for clarity. It is advisable to provide explanations for different splicing categories in Fig. 3C.
5. p.7, last paragraph: It is important to clarify that SF3B1 is a marker for the activated spliceosome complex Bact and not for complex C. The SF3 complex dissociates from the spliceosome during the transition from Bact to B*. Complex B corresponds to the pre-activated spliceosome rather than the

pre-catalytic spliceosome. The same errors in the middle of p.8 and on Fig. 6C need correction.

6. The authors assert that the PRPF8-U4 interaction was not affected by the RP mutation (p. 8), but Fig. 6B shows that there were clearly more hits in two regions of U4 in RO and RPE.

7. p.9, top: The authors mention that exons with increased PRPF8 interaction are rich in thymidine, whereas the downregulated exons are rich in cytidine in ROs. While this is true from the data in Fig. 7E, I noted that even the intron sequences near the 5' splice site (nucleotide positions >0) exhibit a similar trend. I am curious to know if this pattern holds true for sequences from iPSC, Kio or RPE as well.

We would like to submit our revised manuscript titled: " Dysregulation of hBrr2 helicase by PRPF8 disrupts human spliceosome kinetics and 5'-splice site selection, revealing tissue-specific alternative and cryptic splicing defects" to *Nature Communications* for further consideration. We thank the reviewers and the editorial board for providing very useful criticisms that have enhanced the value of our manuscript. We have considered the comments made by all reviewers. All of the new additions and corrections are highlighted in red throughout the main text. The reply to reviewers is show in blue font.

Reviewer #1

Atkinson and colleagues are proposing a manuscript where they delved into the mysteries of a spliceosomal scaffolding protein PRPF8 in which mutations are causative for an inherited retinal degenerative disease, retinitis pigmentosa-type 13. Using a comprehensive combination of RP13 patient-derived iPSCs, and isogenic controls where the mutation had been corrected using CRISPR/Cas9, as well as RPE cells, retina organoids and kidney organoids, they were able to decipher the mechanistic underpinning of RP13 and why PRPF8 mutations cause retina-confined pathology, despite PRPF8 being abundantly expressed in our bodies. I found the manuscript pleasurable to read and the topic interesting. The work can have substantial clinical relevance with respect to therapeutic approaches. I did not find serious flaws in the manuscript, but I have a few major and several minor comments that should be addressed before publication.

Major:

- A typical clinical RP manifests as photoreceptor dysfunction and death whereas the inner retina remains relatively intact, including the retinal ganglion cells (RGCs). Still, the authors show significant loss of the RGC marker SNCG (Fig. 1F) in their RP13 retinal organoids (RO). On the other hand, bipolar, horizontal and amacrine cell markers show similar expression in RP13 and control ROs. Isn't it controversial that RGCs would be so clearly affected in a RP phenotype, and does it affect the translational validity of the model if this is the case?

We thank the reviewer for this important comment. We were surprised to observe the enhanced loss of retinal ganglion cells in RP13-ROs, but would like to reiterate that, clinically, the thinning of both the outer and inner retina is observed (Martin McKibbin, personal communication). Recent published work has shown that variants in the N-terminal of *PRPF8* gene are associated with glaucoma (PMID 28707069). Importantly, in our previous work (PMID 30315276), we have shown that PRPF31 patient-specific retinal organoids display an impaired response to GABA, the most abundant inhibitory neurotransmitter in the retina, which modulates the activity of different types of retinal ganglion cells. These findings may warrant further examination of PRPF8 patients and could impact on the clinical application of future therapies.

- The authors should use the maximum granularity in data presentations. The authors use bar graphs with individual data points, or scatter plot, in e.g. Fig. 4B, but not throughout the manuscript. Using scatter plot, the reader can easily see the sample size and make more informed conclusions themselves. Along these lines, the difference in e.g. Fig. 1G does not look $P < 0.01$ significant to me. Similarly, the effect in 4H does not look $P < 0.0001$. The authors thank the reviewer for their feedback and have presented the data in 1G and 4H as box and whisker plots. This better presents the dispersion and centrality of the data, and makes it clearer why the comparisons have $p < 0.01$ and $p < 0.0001$ significance, respectively. All of the other graphs have now been modified to display all experimental data points.

- The materials & methods part is lengthy (+6000 words), which is understandable because so many techniques have been used. Still, I suggest trying to cut it a bit shorter by e.g. referencing to earlier work. Nat Commun guide to authors say: "Methods should be written as concisely as possible and typically do not exceed 3,000 words but may be

longer if necessary. We encourage you to deposit any step-by-step protocols used in your study in Protocol Exchange, an open resource maintained by NPG.”

We have taken this suggestion into consideration and have shortened the methods section considerably. The detailed methods section is now included in the Supplementary Information.

- Also, some of the figure legends exceed the suggested <350 words, particularly in Fig. 2. Guide to authors also suggest abstract length of approximately 150 words. Current abstract is ~200.

Thank you for this very useful comment. The abstract has been shortened to 150 words and all figure legends are now less than 350 words in length.

Minor:

- In the abstract, “rod degeneration” and “photoreceptor loss” are essentially synonyms. Can remove “rod degeneration”.

Thank you, this has been changed to read “retinal-specific endophenotypes comprising photoreceptor loss...” (revised abstract, page 1).

- There are several flaws in the references list. For example, the authors cite preprints of old papers, although proper bibliography must exist. Citation 7 has two different years?

Thank you, the reference list has been revised thoroughly.

- Before citing Figure S5F, I suggest to edit: “Retinal and photoreceptor degeneration were among the most affected “disease and function” categories”.

Thank you, this sentence is now changed as suggested (revised results, page 4).

- Before citing Fig. 2D, cannot say “significantly larger” if there is no quantification of the phenomenon. Please edit text.

Thank you, this has been changed to “Cajal body clusters were larger...” (revised results, page 5).

- In page 6, I suggest removing the word “Strikingly”. Having seen/read the earlier results, the amount of DEGs was not that striking to me at least.

Thank you, we have removed this adverb as suggested (revised results, page 6).

- In page 10, I am not sure if the current citation 26 is the best. In that paper, the correlation of transcriptomic and proteomic data actually looks relatively strong. I don't see that literature citation here is all that necessary.

Thank you, this reference has now been removed.

- In early discussion, please do a fact check. I think Hartong and Dryja state RP prevalence at 1:4000. Today, many references state 1:3000-1:4000. Also, according to Hartong and Dryja, PRPF mutations may not account for 15-20 % of adRP, but a lower amount at ~ 11 %.

Thank you, this introductory sentence now reads: "RP is one of the most common forms of sight loss with a prevalence of about 1 in 4000 births and more than 1 million people affected worldwide²⁹. Autosomal dominant inheritance accounts for ~ 30-40% of RP, with ~ 11% caused by mutations in PRPFs" (revised discussion, page 10).

- Please soften the statement: "...collectively unravelling the mechanisms of RP pathogenesis". It remains complex and there are so many types of RP.

Thank you: this has been softened to focus on PRPF-RP pathogenesis as follows "unravelling the mechanism of PRPF-RP pathogenesis" (revised discussion, page 10)

- Figure 5A. The authors should show full Western blots as supplementary data.

Thank you, we have compiled images of complete, uncropped western blots and include them in Supplementary Information.

Reviewer #2

Heterozygous mutations that affect certain splicing factors lead to forms of retinitis pigmentosa (RP) in humans. Presently, it is not understood how the resulting defects in ubiquitously required splicing factors lead to a tissue-specific disease phenotype. Atkinson et al. addressed this question with respect to a RP13-linked mutation in the large spliceosomal scaffold, PRPF8. Specifically, they investigated a mutation that leads to a H2309P exchange in PRPF8. The affected residue is positioned at the base of a Jab1/MPN domain at the C-terminus of PRPF8. This domain has been shown to bind the spliceosomal

helicase BRR2 and regulate BRR2 helicase activity, in part via a C-terminal tail that can be inserted into the BRR2 RNA-binding tunnel. To investigate effects of H2309P-PRPF8, the authors reprogrammed fibroblasts with the relevant heterozygous mutation derived from four patients into iPSCs, and differentiated those into retinal pigment epithelium (RPE) cells, retinal organoids (ROs), and, for comparison, kidney organoids (KiOs). They also generated isogenic controls, in which the mutation was corrected by CRISPR/Cas9-based genome engineering. They used image-based cellular and ultrastructural investigations to demonstrate that iPSC-derived retinal tissue exhibits RP disease phenotypes. They then conducted transcriptomics and quantitative proteomics studies, revealing splicing defects as well as abnormal cilia and functional defects in retinal cells and organoids. Splicing defects were confirmed via a lentiviral splicing reporter. The authors also characterized structurally unique splicing clusters in human photoreceptor cells at the nuclear periphery. Using iCLIP-seq, the authors showed that H2309P-PRPF8 exhibited altered binding to 5'-splice sites of transcripts that encode ciliary proteins as well as altered interactions with a key functional region of U6 snRNA and snoRNAs in retinal organoids.

The authors developed a powerful model system to study the tissue-specific effects of a RP-linked mutation affecting a key splicing factor. The battery of powerful state-of-the-art cellular biological, multi-omics and biochemical approaches they employed to characterize disease phenotypes on the tissue, cellular and molecular level is truly impressive. While this reviewer is no expert in the techniques applied, the work appears to be technically sound and the results support the conclusions drawn. This work not only uncovers disease principles of a particular mutation affecting a splicing factor, but also provides a blueprint for delineating how mutations that affect ubiquitously required factors manifest in a tissue-specific manner. Of particular interest is the discovery of a special organization of subnuclear splicing compartments in human photoreceptors. The manuscript is well-written and should be accessible to a wide audience. This reviewer has only very few specific comments/questions:

1. Can the authors discuss in more detail how dysregulation of BRR2 by PRPF8 RP13 variants may affect 5'-splice site selection, as they detect in their data? Specifically, how PRPF8-mediated regulation of BRR2 might affect the selection of suboptimal 5'-splice sites?

It has previously been shown that depletion of hBrr2 reduces selection of optimal 5'SS and enhances the usage of cryptic sites of a reporter gene suggesting a role for Brr2 in

maintaining the fidelity of 5'SS (PMID 2432620). Moreover, RP-linked mutations (S1087L and R1090L) in the hBrr2's RNA-binding tunnel in close proximity to the PRPF8 C-terminal tail that may weaken the tail interaction, were shown to result in a higher splicing efficiency and increased 5'-cryptic splice-site selection of a reporter gene, suggesting a role for hBrr2 in maintaining the 5'-splice site identity (PMID 2432620). A genetic screen in *Caenorhabditis elegans* has identified a strong suppressor of 5'SS mutation and cryptic splicing in the unstructured N-terminal region of Brr2 (PMID 36321655). Moreover, truncation of this region in yeast has resulted in the reduction of U6 and U5, and accumulation of U1 in the B^{act} spliceosome (PMID 25670679). Thus, a timely loading of hBrr2 on U4 and unwinding of U4/U6, upon formation of the 5'SS-ACAGA-box helix, is important for maintaining 5'SS in the catalytic centre of the spliceosome. It is conceivable that premature activation of hBrr2 or its impaired inhibition after U4 release may lead to the interaction of ACAGA-box with a second alternative or cryptic 5'SS, or formation and accumulation of defective B^{act}, respectively. These scenarios are consistent with our observation of increased differential alternative splicing and cryptic splicing in RP13 retinal cells as well as accumulation of the B^{act} complex in these cells. We have included this more complete description in the discussion section of the revised manuscript (pages 11, 12).

2. The authors describe that the H2309P exchange may “destabilize” the PRPF8 C-terminal tail that can intermittently inhibit BRR2, suggesting that H2309P-PRPF8 may inhibit BRR2 less efficiently. Previously, it was shown that RP13-linked variants of the PRPF8 Jab1/MPN domain can have diverse effects on BRR2 helicase activity (RNA Biol 11, 298; 2014). Is it known how a H2309P variant of PRPF8 (or of the Jab1/MPN domain) affects BRR2 activity? Can this effect explain the splicing defects observed?

This mutation is located at the hinge that links the globular region of the Jab1/MPN domain to the tail, and it was previously proposed to disrupt the interaction of Jab1 with hBrr2 and thus affect the hBrr2 helicase activity (PMID 24643059). Alternatively, it could de-stabilise binding of the C-terminal tail to the hBrr2's RNA binding tunnel and result in an increased activity of hBrr2. The equivalent mutation in yeast has reduced binding of the yeast Jab1 domain to γ Brr2 *in vitro* (PMID: 19098916) and yeast cells carrying this mutation displayed a strong temperature sensitivity and *in vivo* splicing defect (PMID 23704370). Our gradient fractionation and Western blotting in RP13-RPE extracts revealed that the PRPF8 binding to hBrr2 in humans is not disrupted by this mutation,

suggesting that the mutant human PRPF8 is viable *in vivo* and maintains its affinity to hBrr2. Previously, two RP-linked mutations (S1087L and R1090L) in the hBrr2's RNA-binding tunnel, in close proximity to the PRPF8 C-terminal tail that may weaken the tail interaction, were shown to result in a shorter interaction time of spliceosomes with pre-mRNA (higher k_{off}) than the wild-type controls in FRAP experiments. Higher splicing efficiency and increased 5'-cryptic splice-site selection of a reporter gene suggested a role for hBrr2 in maintaining the fidelity of 5'-splice site recognition (PMID 24302620). However, these hBrr2 mutants, in isolation, exhibited significantly lower helicase activities than the wild-type hBrr2. Notably, while the U5 proteins PRPF8 and hBrr2 as part of the spliceosome interact with pre-mRNA for ~30 seconds, the release of U4 proteins from the spliceosome by Brr2 is very fast and lasts ~1 second (PMID 20921136). In addition, formation of the 5'SS/U6 helix triggers remodelling of the spliceosome's RNA-protein network to allow hBrr2 relocation to its loading sequence in the U4 snRNA for U4/U6 unwinding. While this process is very fast, its disruption can lead to unproductive activation of the spliceosome (e.g. deletion of the N-terminal domain of yBrr2 results in unproductive dissociation of the tri-snRNP into U4/U6 and U5 by yBrr2 in the presence of ATP). Altogether, these data suggest that the timely activation of hBrr2 and its inhibition after spliceosome activation by the Jab1 domain and its C-terminal tail are important for maintaining the splicing fidelity, which needs to be tightly controlled. We have included part of this more complete description in the discussion section of the revised manuscript (pages 11, 12).

3. Do the authors attribute any disease relevance to the altered snoRNA interactions detected for H2309P-PRPF8?

We thank the reviewer for this comment. We have added a note in the discussion section (page 12) of the revised manuscript: "Apart from mediating post-transcriptional modification of small non-coding RNAs (such as ribosomal RNAs and snRNAs), snoRNAs have been implicated in alternative splicing and micro-RNAs production. Moreover, a fraction of snoRNAs have been found associated with spliceosomes. Although the role of snoRNAs in oncogenesis and autoimmune diseases have been documented, their function in hereditary retinal diseases is currently unclear and remains to be explored. Our data suggests that they may play a role in the etiopathogenesis of PRPF8 RP."

Reviewer #3

Atkinson et al. investigated the impact of pathogenic RP mutations in the PRPF8 gene on splicing in humans. They utilized iPSCs-derived retinal organoids (RO), retinal pigment epithelium (RPE) and clinically unaffected kidney organoids (KiOs) from patients carrying PRPF8 RP type 13 c.6926A>C (p.H2309P) heterozygous missense mutation, and their isogenic controls generated by correcting the mutation using CRISPR/Cas techniques. They then performed a comprehensive set of analyses, including imaging, RNA-cross-linking, transcriptomic and proteomic assessment to investigate differences in splicing control between these cells. Their results reveal impairment of alternative splicing predominantly in ciliary and retinal-specific transcripts.

PRPF8 plays a pivotal role in the spliceosome function, orchestrating the catalytic core of the spliceosome through interactions with pre-mRNA, U2, U5, U6 snRNAs, and various protein components. In yeast, the C-terminus of Prp8 is known to regulate the ATPase and RNA helicase activities of Brr2, which are essential for spliceosome activation. This specific region is susceptible to RP mutations that affect the interaction between Prp8 and Brr2. However, the mechanism responsible for the tissue-specific effects of these mutations remains unknown. In this study, the authors employed a range of analytical methods to investigate the impact of RP mutations on the splicing process in humans. They demonstrate that RP mutations can affect PRPF8-U6 interaction, the splicing efficiency, splicing kinetics, and the organization of nuclear speckles. Specifically, they have a tissue-specific impact on alternative splicing in 5' splice sites selection. While the precise mechanism driving the tissue specificity of these mutations remains unresolved, it represents a significant advancement in our efforts to understand how these mutations give rise to RP phenotypes.

This work involves an extensive analysis of genome-wide data, but there are instances where the interpretations of data lack clarity. To make this work more accessible to broader readers, it would be beneficial for the authors to provide clear explanations. Moreover, the finding that RP mutations impair 5' splice site recognition is in line with the current notion that the 5' splice site is freed from U1 binding for interaction with U6 before the action of Brr2. It may be worthwhile to incorporate this concept into the discussion. Specific comments addressing inaccuracies in the use of certain terminology

and in the description of the outlined mechanisms are detailed below.

...the finding that RP mutations impair 5' splice site recognition is in line with the current notion that the 5' splice site is freed from U1 binding for interaction with U6 before the action of Brr2. It may be worthwhile to incorporate this concept into the discussion.

Thank you, we have incorporated this concept in the discussion section on pages 11 and 12.

Specific comments addressing inaccuracies in the use of certain terminology and in the description of the outlined mechanisms are detailed below.

1. p.3, line 7, "Fine-tuning of Brr2 activity may modulate splice site proofreading....". Using the term "splice site selection" is more suitable than "proofreading". There is no evidence for splice site proofreading in any known human cases. The prevalence of alternative splicing, which accommodates a wide array of splice site sequences, adds complexity to the concept of proofreading.

We thank the reviewer for this comment and have used the term "splice site selection" instead of proofreading in the revised version of the manuscript.

2. Fig. 2C: It is not apparent to me that the amount of U4 is reduced in RPE as the intensities of U5 and U6 decreased correspondingly. This may be attributed to the possibility of lighter exposure in the RPE blot compared to the control. Furthermore, several bands observed in fractions 1-5 of the control that are absent in RPE sample, which need further explanation or discussion.

We have now added a quantification of the Northern blot bands in the supplementary Figure S7E. In addition, a note has been added to the figure legend 2C.

3. Figure 2F shows that the PRPF8 p.H2309P mutation had a minor impact on splicing efficiency in iPSC and KiO cells, while it had a more pronounced effect on RPE and RO cells, albeit in opposing directions. The authors should provide an explanation for this divergence between RPE and RO cells. It is conceivable that distinct introns may be regulated differently, given that only one specific intron was examined in this experiment. To draw a conclusion, the authors should replicate this experiment with other introns.

The construct used to evaluate the splicing efficiency contains a strong chimeric globin-immunoglobulin intron and has been designed to shorten the luciferase mRNA and protein half-lives to be more sensitive to splicing changes (PMID 20123975, page 5 of revised manuscript). Moreover, we measured both the intronless luciferase and intron-containing luciferase signals to control for possible changes in transduction efficiencies between patient and control cells. In general, splicing efficiency is thought to be determined by sequence features, transcription level and concentration of splicing machinery. Recent studies have demonstrated that transcripts localised near the nuclear speckles have higher splicing efficiency due to a higher concentration of spliceosomes (PMID 31133700). Moreover, the gene organisation around the nuclear speckles is dynamic and can be different among various cell types. We speculate that the special organisation of splicing clusters in human photoreceptor cells may drive higher splicing efficiency in these cells that is consistent with the characteristics of retina, including larger amounts of snRNAs and spliced housekeeping genes, as well as a higher number of alternative splicing events, compared to other tissues. A thorough analysis of this hypothesis requires further study using introns with different splice sites strength and mutated splice sites as well as quantitative FISH to localise the introns and mRNA in photoreceptors, which is beyond the remit of the current study. Nonetheless, our comprehensive bulk and single cell transcriptomic analyses combined with iCLIP-seq represent the most detailed currently available profiles of constitutive and alternative splicing changes in patient-derived cell types.

4. Fig. 3: The abbreviations used, such as BP, CC, MF in 3BCF, and A3SS, A5SS, MXE ... etc. in 3E, should be defined in the figure legends for clarity. It is advisable to provide explanations for different splicing categories in Fig. 3C.

This has been amended (revised figure legends, page 31)

5. p.7, last paragraph: It is important to clarify that SF3B1 is a marker for the activated spliceosome complex Bact and not for complex C. The SF3 complex dissociates from the spliceosome during the transition from Bact to B*. Complex B corresponds to the pre-activated spliceosome rather than the pre-catalytic spliceosome. The same errors in the middle of p.8 and on Fig. 6C need correction.

We are grateful to the reviewer for this comment. We have now corrected this error in the revised version of the manuscript and Figure 6C, and have changed pre-catalytic to “pre-activated” spliceosome.

6. The authors assert that the PRPF8-U4 interaction was not affected by the RP mutation (p. 8), but Fig. 6B shows that there were clearly more hits in in two regions of U4 in RO and RPE.

We thank the reviewer for the comment. Indeed, the total numbers of reads on U4 over nucleotides 36-96 increased in RP13 retinal cells compared with the controls, although this effect was not reproducible across biological replicates. However, we did not detect deletions in U4 sequencing reads, suggesting that the U4 binding might be indirect via other proteins (e.g. hBrr2) or U4 base pairing with U6, which is not disrupted in the iCLIP experiments. In the revised version of the manuscript, we have indicated this in the results section (page 8).

7. p.9, top: The authors mention that exons with increased PRPF8 interaction are rich in thymidine, whereas the downregulated exons are rich in cytidine in ROs. While this is true from the data in Fig. 7E, I noted that even the intron sequences near the 5' splice site (nucleotide positions >0) exhibit a similar trend. I am curious to know if this pattern holds true for sequences from iPSC, Kio or RPE as well.

We have provided this information in the revised Figure S9F and have added one sentence “however this was not the case for RP13-RPE, KiOs or iPSCs (Figure S9F)....” (revised results, page 9)

We hope that these revisions are satisfactory. We look forward to hearing from you in due course.

Sincerely (on behalf of all senior authors),

Majlinda Lako, PhD

Professor of Stem Cell Science

Reviewers' Comments:

Reviewer #1:

Remarks to the Author:

Thank you. My critique have been adequately addressed.

Reviewer #2:

Remarks to the Author:

In revising their manuscript, the authors have adequately addressed all points raised by this reviewer.

Reviewer #3:

Remarks to the Author:

The authors have appropriately addressed the prior comments, and the current version explains more clearly how the RP-linked PRPF8 mutations may impact 5'SS selection. I am satisfied with the revision and recommend publication of the manuscript. A typo on page 12, line 473: "hBr2" should be corrected to "hBrr2".

We would like to submit our revised manuscript titled: " PRPF8-mediated dysregulation of hBrr2 helicase disrupts human spliceosome kinetics and 5'-splice-site selection causing tissue-specific defects" to *Nature Communications* for further consideration. We thank the reviewers and the editorial board for providing very useful criticisms that have enhanced the value of our manuscript. We have considered the comments made by all reviewers. All of the new additions and corrections are highlighted in red throughout the main text. The reply to reviewers is show in blue font.

Reviewer #1 (Remarks to the Author):

Thank you. My critique have been adequately addressed.

Reviewer #2 (Remarks to the Author):

In revising their manuscript, the authors have adequately addressed all points raised by this reviewer.

Reviewer #3 (Remarks to the Author):

The authors have appropriately addressed the prior comments, and the current version explains more clearly how the RP-linked PRPF8 mutations may impact 5'SS selection. I am satisfied with the revision and recommend publication of the manuscript. A typo on page 12, line 473: "hBr2" should be corrected to "hBrr2".

We thank the reviewer for spotting this type which has now been corrected.

We hope that these revisions are satisfactory. We look forward to hearing from you in due course.